# Fog signaling has diverse roles in epithelial morphogenesis in insects

**Matthew Alan Benton[1,2†‡], Nadine Frey[1†], Rodrigo Nunes da Fonseca[1§#], Cornelia von Levetzow[1¶], Dominik Stappert[1**], Muhammad Salim Hakeemi[1], Kai H Conrads[1], Matthias Pechmann[1], Kristen A Panfilio[1,3], Jeremy A Lynch[4], Siegfried Roth[1]***

[1]Institute for Zoology/Developmental Biology, Biocenter, University of Cologne, Köln, Germany; [2]Department of Zoology, University of Cambridge, Cambridge, United Kingdom; [3]School of Life Sciences, University of Warwick, Coventry, United Kingdom; [4]Department of Biological Sciences, University of Illinois, Chicago, United States

**\*For correspondence:**
siegfried.roth@uni-koeln.de

[†]These authors contributed equally to this work

**Present address:** [‡]Department of Zoology, University of Cambridge, Cambridge, United Kingdom; [§]Instituto Nacional de Ciência e Tecnologia em Entomologia Molecular (INCT-EM), Centro de Ciências da Saúde, Rio de Janeiro, Brazil; [#]Laboratório Integrado de Ciências Morfofuncionais (LICM), Instituto de Biodiversidade e Sustentabilidade (NUPEM), Universidade Federal do Rio de Janeiro (UFRJ), Macaé, Brazil; [¶]Centrum für Interierte Onkologie, Universitätsklinikum Köln, Köln, Germany; [**]German Center for Neurodegenerative Diseases, Bonn, Germany

**Competing interests:** The authors declare that no competing interests exist.

**Abstract** The *Drosophila* Fog pathway represents one of the best-understood signaling cascades controlling epithelial morphogenesis. During gastrulation, Fog induces apical cell constrictions that drive the invagination of mesoderm and posterior gut primordia. The cellular mechanisms underlying primordia internalization vary greatly among insects and recent work has suggested that Fog signaling is specific to the fast mode of gastrulation found in some flies. On the contrary, here we show in the beetle *Tribolium*, whose development is broadly representative for insects, that Fog has multiple morphogenetic functions. It modulates mesoderm internalization and controls a massive posterior infolding involved in gut and extraembryonic development. In addition, Fog signaling affects blastoderm cellularization, primordial germ cell positioning, and cuboidal-to-squamous cell shape transitions in the extraembryonic serosa. Comparative analyses with two other distantly related insect species reveals that Fog's role during cellularization is widely conserved and therefore might represent the ancestral function of the pathway.
DOI: https://doi.org/10.7554/eLife.47346.001

## Introduction

The Folded gastrulation (Fog) pathway is one of the few signaling pathways dedicated to epithelial morphogenesis (*Gilmour et al., 2017*; *Manning and Rogers, 2014*). Fog signaling was discovered in the fly *Drosophila melanogaster,* where it is required for the formation of two major epithelial folds during early embryogenesis: the ventral furrow, and the posterior gut fold (*Costa et al., 1994*; *Parks and Wieschaus, 1991*; *Sweeton et al., 1991*; *Zusman and Wieschaus, 1985*). The ventral furrow leads to the internalization of the mesoderm, while the posterior gut fold leads to internalization of the hindgut and posterior endoderm (*Campos-Ortega and Hartenstein, 1997*). These folds are formed by coordinated changes in cell shape that are driven by the reorganization of cytoskeleton components and the remodeling of cell junctions. Crucially, the inward directionality of the folding is caused by constrictions of the cells at their apical side, and it is this process that is coordinated by Fog signaling (*Dawes-Hoang et al., 2005*; *Kölsch et al., 2007*; *Martin et al., 2009*).

The molecular basis of the Fog signaling pathway has been extensively studied in *Drosophila*. Fog itself is an extracellular ligand that is secreted by future invaginating cells (*Dawes-Hoang et al., 2005*) and activates two G protein-coupled receptors (GPCRs): Mist (Mesoderm-invagination signal transducer, also known as Mthl1 [Methuselah-like1]) (*Manning et al., 2013*) and Smog (*Jha et al., 2018*; *Kerridge et al., 2016*). Activation of these receptors causes Concertina (Cta), the Gα12/13 subunit of a trimeric G protein, to recruit RhoGEF2 to the apical plasma membrane, where it

promotes myosin II contractility (via Rho and Rho kinase), thereby triggering apical cell constrictions (*Barrett et al., 1997*; *Dawes-Hoang et al., 2005*; *Kölsch et al., 2007*) (*Figure 1—figure supplement 1*).

Although Fog is a secreted ligand, it appears to only act locally (*Costa et al., 1994*; *Dawes-Hoang et al., 2005*; *Bailles et al., 2019*). Because of this, the localized expression of *fog* and *mist* in the presumptive mesoderm and posterior endoderm provides the spatial specificity of the pathway (*Costa et al., 1994*; *Manning et al., 2013*).

It is important to note that in the absence of Fog signaling, some cells do still undergo apical constriction in the ventral furrow and posterior gut fold. However, fewer cells constrict, and the spatial and temporal coordination of those constrictions is disrupted. As such, Fog signaling is proposed to promote and coordinate apical constriction of cells across invaginating epithelia (*Costa et al., 1994*; *Sweeton et al., 1991*). Specifically, in *fog* mutants alone, ventral furrow formation is irregular and delayed compared with wildtype, but mesoderm internalization still occurs (*Parks and Wieschaus, 1991*; *Seher et al., 2007*; *Sweeton et al., 1991*). The transmembrane protein T48 also recruits Rho-GEF2 apically and induces apical constrictions, in a Fog-independent manner (*Kölsch et al., 2007*) (*Figure 1—figure supplement 1*). Only deletion of both *fog* and *T48* prevents mesoderm internalization (*Kölsch et al., 2007*). In contrast, posterior gut folding and endoderm internalization are completely dependent on Fog signaling (*Seher et al., 2007*; *Zusman and Wieschaus, 1985*). Here, Fog fulfills two functions: it induces apical constrictions locally, and it triggers a directional wave of Rho/MyoII activation that drives the propagation of cell invaginations outside of (anterior to) the *fog* expression domain (*Bailles et al., 2019*).

The Fog pathway is also involved in other morphogenetic events. During late embryogenesis, it is required during salivary gland morphogenesis and it affects the folding of imaginal discs in larvae (*Chung et al., 2017*; *Manning et al., 2013*; *Nikolaidou and Barrett, 2004*). Most recently, loss of Fog signaling was found to affect cell intercalation during germband extension (*Jha et al., 2018*; *Kerridge et al., 2016*), thus revealing functions for Fog signaling independent of apical constrictions.

The importance of Fog signaling during development in other insects is largely unknown. While the pathway components have been identified in many lineages, the morphogenetic basis of early development greatly varies between different species (*Anderson, 1972a*; *Anderson, 1972b*; *Roth, 2004*; *Urbansky et al., 2016*).

Recent molecular analysis in the midge *Chironomus riparius* has also cast doubts about the functional conservation of the pathway for early embryonic development. Rather than forming a highly stereotyped ventral furrow, *Chironomus* embryos internalize their mesoderm via cell ingression, and this event is only weakly affected by loss of Fog signaling (*Urbansky et al., 2016*). However, overexpression of *fog* and/or *T48* causes the formation of a ventral furrow and invagination of mesoderm in a *Drosophila*-like mode. Based on their results, *Urbansky et al. (2016)* hypothesized that Fog signaling was recruited from a later role in development to an early role in gastrulation in the *Drosophila* lineage. However, as pointed out by the authors, an alternative hypothesis is that Fog signaling has a more widely conserved role in early development and this has been reduced in the lineage leading to *Chironomus.*

To test whether Fog signaling does have a more widely conserved role in early development, we have analyzed Fog pathway components in the beetle *Tribolium castaneum*. In contrast to *Drosophila melanogaster* and other dipteran species like *Chironomus riparius*, many features of *Tribolium* embryogenesis are more typical of insects in general, including the mechanism and timing of blastoderm cellularization (*van der Zee et al., 2015*), the mode of germ cell formation (*Schröder, 2006*), germband formation (*Benton, 2018*), extraembryonic tissue development (*Hilbrant et al., 2016*; *Horn and Panfilio, 2016*; *van der Zee et al., 2005*) and segmentation (*Clark and Peel, 2018*; *Sommer and Tautz, 1993*). Therefore, analyzing Fog signaling in *Tribolium* will reveal the role of the pathway within a developmental context that is more representative of insects.

Our analysis of Fog signaling in *Tribolium* reveals that, in contrast to *Chironomus* but like in *Drosophila*, this pathway contributes to mesoderm internalization and drives an early invagination at the posterior pole. In addition, *Tribolium* Fog signaling is involved in several aspects of development that have been lost or modified in the lineage leading to *Drosophila,* such as the extensive epithelial folding that leads to germband formation, the simultaneous spreading of the extraembryonic serosa, the apical-basal positioning of primordial germ cells, and even the cellularization of the blastoderm.

The latter function is deeply conserved, as we also observed it in bug and cricket embryos representing distant branches of the insect phylogenetic tree.

## Results

### *Tc-cta*, *Tc-mist* and *Tc-fog* are expressed in morphogenetically active tissues

As a first step towards characterizing the Fog signaling pathway in *Tribolium*, we identified and cloned the pathway components (*Figure 1—figure supplement 1*) and characterized their expression during development. The *Tribolium* genome contains one ortholog each for *fog, mist* and *cta* (hereafter referred to as *Tc-fog, Tc-mist* and *Tc-cta*). Fog is a fast evolving protein with very low overall sequence conservation within insects and no detectable homologs in currently available non-insect genomes (*Figure 1—figure supplement 2*) (*Urbansky et al., 2016*). In contrast, previous research has shown mist and cta to be well conserved among insects (*de Mendoza et al., 2016*; *Kozasa et al., 2011*; *Manning et al., 2013*; *Parks and Wieschaus, 1991*; *Urbansky et al., 2016*).

In *Drosophila, fog*, *mist* and *cta* are all maternally expressed (*Costa et al., 1994*; *Manning et al., 2013*; *Parks and Wieschaus, 1991*; *Urbansky et al., 2016*). In *Tribolium*, we were unable to detect maternal expression for any of the three genes using conventional whole-mount RNA in-situ hybridization (ISH) (for staging and description of wildtype development see *Benton et al., 2013*; *Handel et al., 2000*). However, publically available RNA-sequencing data revealed the presence of maternal transcripts for *Tc-cta* and *Tc-fog* (*Dönitz et al., 2018*).

After blastoderm formation, *Tc-cta* and *Tc-mist* transcripts were uniformly distributed, while *Tc-fog* transcripts were enriched at the anterior pole (*Figure 1—figure supplement 3A,I,Q*). During later blastoderm stages, *Tc-cta* formed a shallow gradient with higher levels towards the posterior

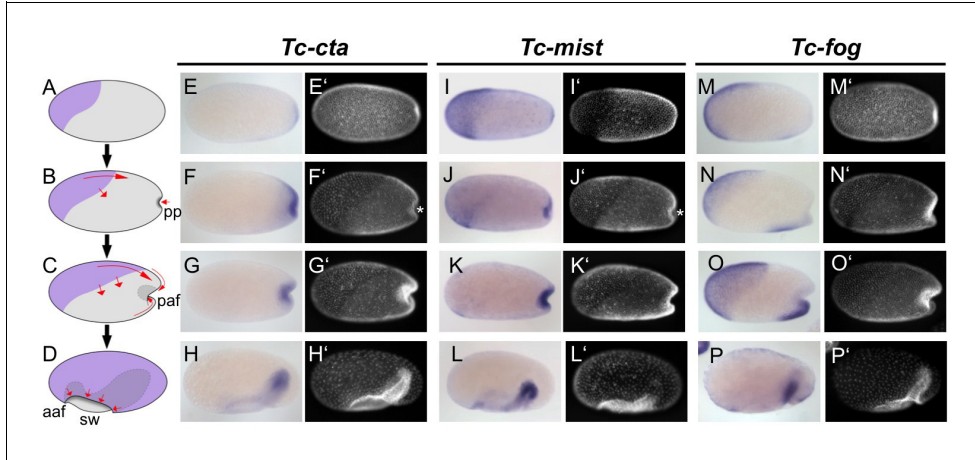

**Figure 1.** Expression of Fog signaling components during early embryogenesis. (A–D) Schematics showing embryo condensation as described in the text. Serosa is shown in purple, germ rudiment tissue is shown in gray, arrows display tissue movements. aaf: anterior amniotic fold, paf: posterior amniotic fold, pp: primitive pit, sw: serosal window. (E–P') Whole mount ISH and DNA staining for *Tc-cta* (E–H), *Tc-mist* (I–L) and *Tc-fog* (M–P). (E'–P') nuclear (DAPI) staining of respective embryos. Anterior is left, ventral is down (where possible to discern).
DOI: https://doi.org/10.7554/eLife.47346.002

The following figure supplements are available for figure 1:

**Figure supplement 1.** Fog and T48 pathway in *Drosophila*.
DOI: https://doi.org/10.7554/eLife.47346.003

**Figure supplement 2.** Insect Fog proteins.
DOI: https://doi.org/10.7554/eLife.47346.004

**Figure supplement 3.** Expression of *Tc-cta*, *Tc-mist* and *Tc-fog* during early embryogenesis.
DOI: https://doi.org/10.7554/eLife.47346.005

**Figure supplement 4.** *Tc-fog* and *Tc-twi* are co-expressed only within the posterior presumptive mesoderm.
DOI: https://doi.org/10.7554/eLife.47346.006

pole (*Figure 1E*), while *Tc-mist* and *Tc-fog* were strongly expressed in an oblique anterior-dorsal domain (the future serosa; *Figure 1A,I,M*). At the same time, weak *Tc-mist* expression became visible at the posterior pole, and weak *Tc-fog* expression was also visible in a patch of cells at the ventral side of the embryo (*Figure 1I,M*; *Figure 1—figure supplement 4B*).

The complex morphogenetic events that transform the *Tribolium* blastoderm into the germband (schematic in *Figure 1A–D*, wildtype in *Videos 1* and *2*) have been described in detail elsewhere (*Handel et al., 2000*; *Benton et al., 2013*), but we will briefly present them here for the benefit of the reader. After cellularization is complete, cells that will form the embryo proper and the extraembryonic amnion (together termed the germ rudiment) undergo mitosis and condense towards the posterior-ventral region of the embryo (*Figure 1B,C*). At the same time, a patch of cells at the posterior pole undergoes apical constrictions to form a cup-shaped indentation (termed the primitive pit, pp in *Figure 1B*) that then deepens into a fold (termed the posterior amniotic fold, paf in *Figure 1C*, *Video 3*). Cells constrict in a pulsatile manner (*Video 4* showing close up of cells), as previously described for the *Drosophila* ventral furrow (*Martin et al., 2009*). Subsequently, the dorsal 'lip' of the posterior amniotic fold moves ventrally, progresses over the posterior pole while undergoing involution, and then moves anteriorly over the ventral face of the embryo. Differences in the relative timing of posterior folding, cell division and tissue condensation between embryos lead to high variability in the overall appearance and progression of the posterior fold (*Video 5* showing four embryos from posterior). As this process occurs, the edges of the posterior amniotic fold spread anteriorly until they meet with the anterior amniotic fold (which forms independently, aaf in *Figure 1D*). During the above condensation and tissue folding, the presumptive serosa cells undergo a cuboidal-to-squamous transition and spread over the entire egg surface without any cell division (*Video 6* showing wildtype serosa flattening). The boundary between serosa and germ rudiment is demarcated by a supracellular actin cable (sca) that may be involved in serosal window closure (sw in *Figure 2D* and wild type in *Video 2*) (*Benton et al., 2013*). Throughout this period, mesoderm internalization occurs along the ventral part of the germ rudiment via both cell ingression and furrow formation mediated by apical constriction (*Handel et al., 2005*).

Throughout embryo condensation, *Tc-cta* expression persisted in the posterior region of the germ rudiment/germ band (*Figure 1F–H*, *Figure 1—figure supplement 3F–H*). *Tc-mist* expression faded first from the dorsal serosal cells, then from the entire serosa, while expression in the primitive pit region/posterior end of the germ band strengthened (*Figure 1J–L*, *Figure 1—figure supplement 3N–P*). *Tc-fog* expression remained in the serosa throughout condensation and became upregulated in a posterior-ventral stripe of cells fated to become mesoderm (*Figure 1N–P*, *Figure 1—figure supplement 3T–X*; *Figure 1—figure supplement 4C–E*). Towards the end of embryo condensation, *Tc-fog* also became expressed in the ectoderm on either side of the mesoderm domain (*Figure 1—figure supplement 3W*).

Our expression analysis shows that *Tribolium* Fog signaling components are activated in a spatiotemporal pattern suggestive of a role in epithelial morphogenesis.

## The Fog pathway is required for the posterior amniotic fold in *Tribolium*

To test whether the Fog signaling pathway is involved in early *Tribolium* embryogenesis, we disrupted *Tc-cta*, *Tc-mist* or *Tc-fog* function via parental RNAi (pRNAi) knockdown (KD) and analyzed both live and fixed embryos.

KD of each of the genes resulted in the same overall phenotype (*Figure 2—figure*

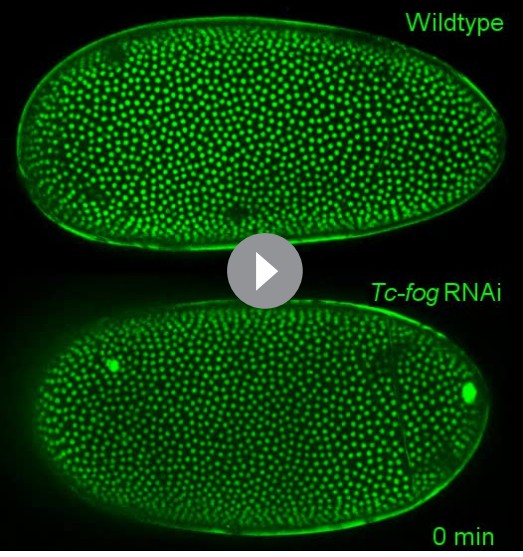

**Video 1.** Fluorescent live imaging of wildtype and *Tc-fog* RNAi nGFP transgenic embryos. Maximum intensity projections of one egg hemisphere are shown with anterior to the left and ventral to the bottom.
DOI: https://doi.org/10.7554/eLife.47346.007

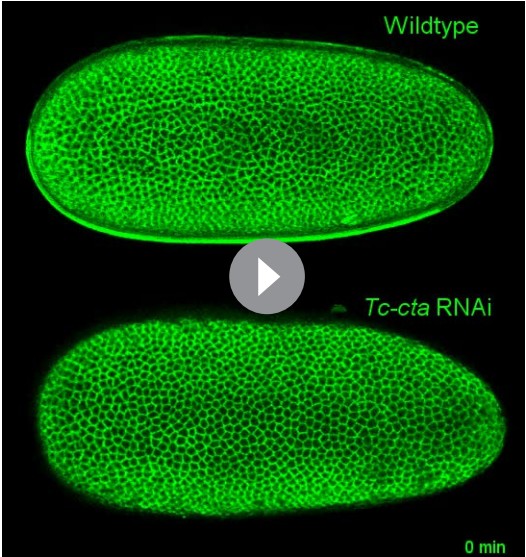

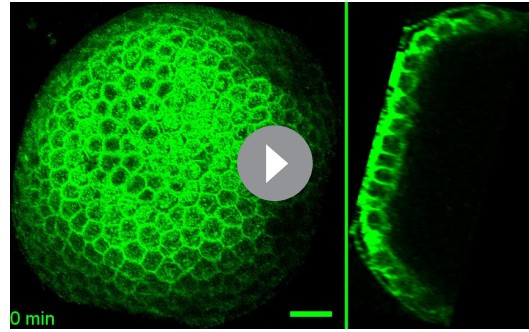

**Video 2.** Fluorescent live imaging of wildtype and *Tc-cta* RNAi LifeAct-GFP transgenic embryos. Maximum intensity projections of one egg hemisphere are shown with anterior to the left and ventral to the bottom.
DOI: https://doi.org/10.7554/eLife.47346.008

**Video 3.** Fluorescent live imaging of the posterior pole of a wildtype embryo transiently expressing GAP43-YFP. Apical constrictions are visible at the center of the forming fold. Embryo was mounted with the posterior pole towards the objective and the resulting movie was digitally rotated. Maximum intensity projection of posterior view is shown as well as a transverse section along the dorsal/ventral midline. Ventral is to the bottom.
DOI: https://doi.org/10.7554/eLife.47346.009

supplement 1). We utilized fluorescent live imaging to better understand the underlying defects (*Figure 2A–H*; *Videos 1*, *2* and *7*). The earliest and most prominent defect was the suppression of primitive pit and posterior amniotic fold formation (*Figure 2B,C,F,G*; *Figure 2—figure supplements 2* and *4*; *Videos 2* and *7*). Because of this lack of folding, the dorsal half of the germ rudiment remained at the dorsal side of the egg in KD embryos (*Figure 2H,K*; *Figure 2—figure supplements 1* and *2*; *Figure 3*).

To investigate whether patterning of the dorsal half of the germ rudiment was disrupted in KD embryos, we analyzed the expression of two known marker genes. Despite the abnormal position of the relevant tissue, both *Tc-pnr* and *Tc-iro*, which are expressed in broad dorsal domains, appeared to be expressed normally in KD embryos (*Figure 2J,K* and *Figure 2—figure supplement 2A–C*). This finding is supported by the fact that a supracellular-actin cable formed between the serosa and germ rudiment tissues, as occurs in wildtype embryos at the same stage (sca in *Figure 2D,H*) (*Benton et al., 2013*).

In addition to the above defects, epithelial holes formed at the serosa/germ rudiment boundary and, during later stages of development, at posterior-ventral regions of the germband (*Figure 2—figure supplements 3* and *4*). In contrast to the major morphogenetic defects in the posterior of the embryo, anterior amniotic fold formation and head condensation still occurred in KD embryos (aaf in *Figure 2H*; *Figure 2—figure supplement 1*; *Figure 2—figure supplement 2D, E* and *Figures 3* and *4* ).

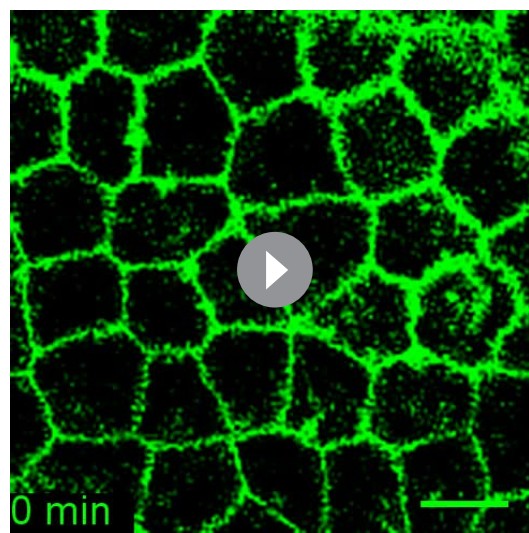

**Video 4.** Single section through the subapical region of the cells at the posterior pole of a wildtype embryo transiently expressing GAP43-YFP. Cells constrict over time and this occurs in a pulsatile manner, and cell intercalation is also visible. Ventral is to the bottom.
DOI: https://doi.org/10.7554/eLife.47346.010

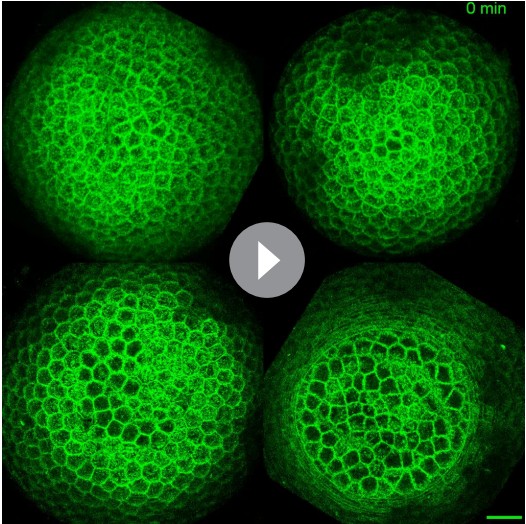

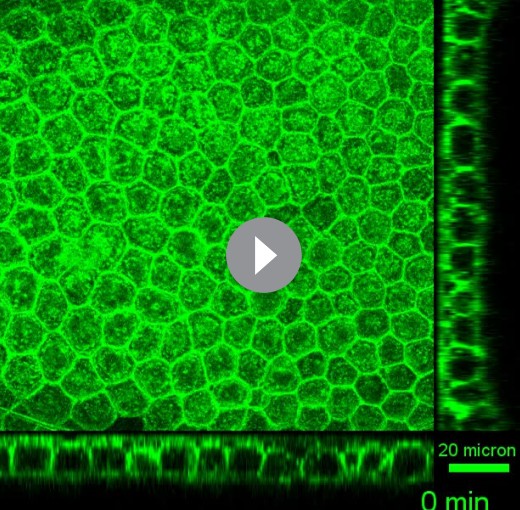

**Video 5.** Fluorescent live imaging of wildtype embryos transiently expressing GAP43-YFP. Embryos were mounted with their posterior poles towards the objective. Maximum intensity projection of posterior view is shown. Ventral is to the bottom.
DOI: https://doi.org/10.7554/eLife.47346.011

**Video 6.** Fluorescent live imaging of the presumptive serosa and germ rudiment epithelium of a wildtype embryo transiently expressing GAP43-YFP. The cuboidal-to-squamous transition of the serosa cells during germband formation can be seen. Maximum intensity projection of the epithelium as well as transverse and sagittal sections along midpoints of the frame are shown. Anterior is to the left, ventral to the bottom.
DOI: https://doi.org/10.7554/eLife.47346.012

Despite the severe changes in overall embryo topology, segmentation was not severely affected in KD embryos. In wildtype embryos, segmentation genes are often expressed in rings that completely encircle the epithelium of the germband (*Benton et al., 2016*; *Sarrazin et al., 2012*). In our KD embryos, these rings were readily visible due to the outward facing topology of the germband (e.g. *Tc-gooseberry* (*Tc-gsb*) (*Davis et al., 2001*) (*Figure 2L,M*).

During germband retraction, KD embryos become highly distorted. However, segments could still be distinguished (*Figure 2—figure supplement 5C,D*). A conspicuous difference to control embryos is the wrong positioning of the hindgut. Instead of pointing inside the embryo, the hindgut points outwards, forming an external tube connected to the posterior tip of the embryo (*Figure 2—figure supplement 5B, D*). This suggests that primitive pit and posterior amniotic fold formation are also required for the correct internalization of the gut.

Taken together, Fog signaling is required for one of the most prominent morphogenetic movements during *Tribolium* gastrulation: the formation of the posterior amniotic fold that is essential both for extraembryonic membrane and gut development. In the absence of Fog the *Tribolium* embryo assumes a topology that is more like that of *Drosophila*: the dorsal ectoderm and extraembryonic tissues remain in a dorsal position (*Figure 3*).

## Fog signaling controls the positioning of the primordial germ cells

*Drosophila* primordial germ cells (PGCs) are specified at the posterior pole of the early embryo and form as 'pole cells' above the surface of the blastoderm (*Cinalli and Lehmann, 2013*). In *Tribolium*, the PGCs are also specified at the posterior of the blastoderm, but they are integrated in the blastoderm cell layer and internalize beneath the blastoderm epithelium at around the same time as primitive pit formation (*Schröder, 2006*). In our live imaging analysis of KD embryos, we frequently observed a posterior ball of tissue (*Figure 2G,H*, white asterisk) and asked whether this tissue consisted of incorrectly localized PGCs.

To follow the development of *Tribolium* PGCs, we examined the expression of the gene *Tc-tapas*, which encodes a Tudor domain protein (*Patil et al., 2014*). *Tc-tapas* has a similar but more robust expression profile than the previously described PGC marker gene *Tc-vasa* (*Schröder, 2006*). In

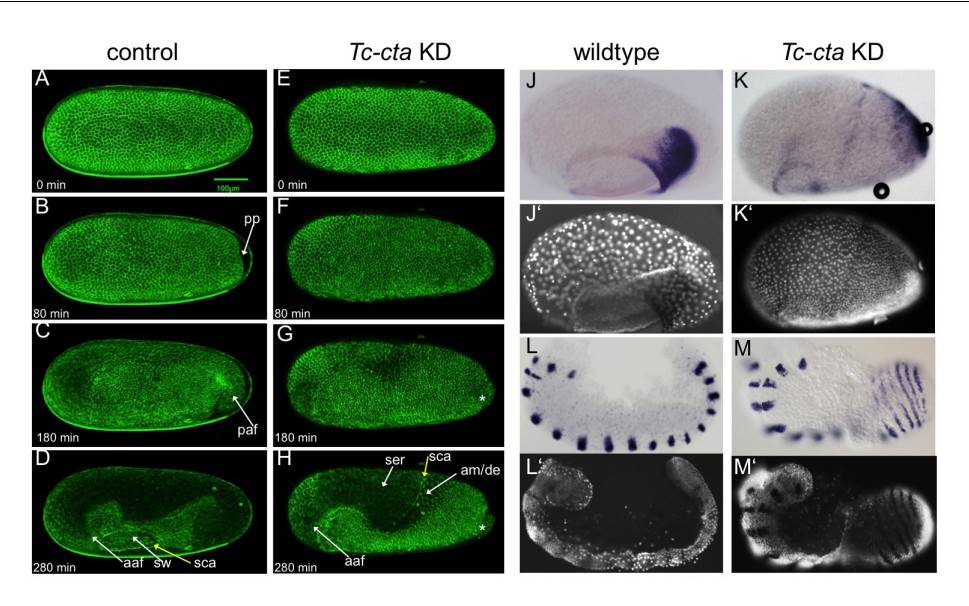

**Figure 2.** Fog signaling is required for posterior amniotic fold formation. (**A–H**) Stills from live fluorescent imaging of LifeAct-eGFP transgenic embryos, ranging from late blastoderm to germband extension stages. (**A–D**) wildtype control. (**E–H**) *Tc-cta* KD. The asterisk marks a cluster of cells that becomes visible at the posterior pole. (**J, K**) *Tc-pnr* is expressed in a broad dorsal domain. (**J', K'**) nuclear (DAPI) staining of respective embryos. (**L, M**) *Tc-gsb* expression marks forming and differentiating segments in elongating germ band embryos. (**L', M'**) nuclear (DAPI) staining of respective embryos. (**J, L**) Wildtype. (**K, M**) *Tc-cta* KD. aaf: anterior amniotic fold, am/de: amnion dorsal ectoderm, paf: posterior amniotic fold, pp: primitive pit, sca: supracellular actin cable, sw: serosal window. Anterior is left, ventral is down.

DOI: https://doi.org/10.7554/eLife.47346.013

The following figure supplements are available for figure 2:

**Figure supplement 1.** KD of *Tc-fog*, *Tc-mist* and *Tc-cta* by RNAi results in similar phenotypes.
DOI: https://doi.org/10.7554/eLife.47346.014

**Figure supplement 2.** Fog signaling is required for posterior amniotic fold formation.
DOI: https://doi.org/10.7554/eLife.47346.015

**Figure supplement 3.** Morphogenetic defects in late *Tc-cta* KD embryos.
DOI: https://doi.org/10.7554/eLife.47346.016

**Figure supplement 4.** Frequencies of phenotypic defects upon *Tc-fog* KD.
DOI: https://doi.org/10.7554/eLife.47346.017

**Figure supplement 5.** Late development of *Tc-cta* KD embryos.
DOI: https://doi.org/10.7554/eLife.47346.018

control embryos, the *Tc-tapas* expressing cells are located in the center of the forming primitive pit (*Figure 4A,D*). During early posterior amniotic fold formation, they leave the epithelium at its basal side by an unknown mechanism. Subsequently, the PGCs form a spherical cluster of cells that remains attached to the posterior end of the segment addition zone (SAZ) during germband extension (*Figure 4B,E*).

In *Tc-fog* pRNAi embryos, *Tc-tapas* was also expressed in a distinct cluster of putative PGCs at the posterior, but in most embryos (85% of KD embryos that displayed phenotypic defects [N = 99], *Figure 2—figure supplement 4*), the cell cluster was located at the apical side of the

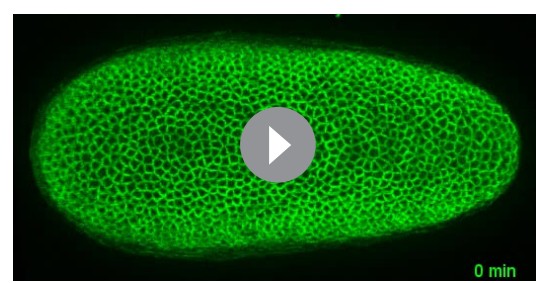

**Video 7.** Fluorescent live imaging of a*Tc-cta* RNAi LifeAct-GFP transgenic embryo. Maximum intensity projection of one egg hemisphere is shown with anterior to the left and ventral to the bottom.
DOI: https://doi.org/10.7554/eLife.47346.020

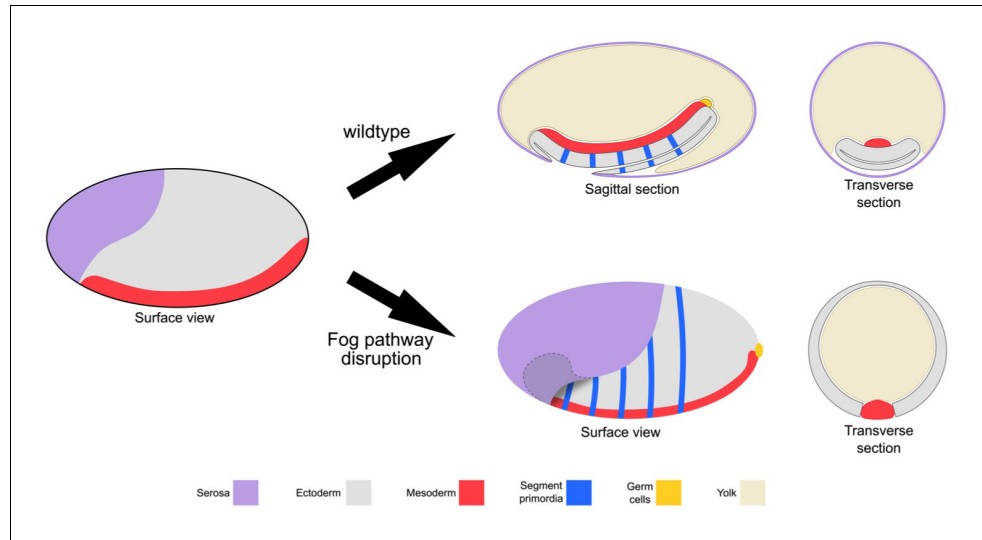

**Figure 3.** Schematic representation of the embryonic phenotype produced by Fog pathway disruption.
Schematics showing wildtype development and the effects on embryo formation of RNAi disruption of *Tc-fog, Tc-mist* or *Tc-cta*. Anterior is left, ventral is down.
DOI: https://doi.org/10.7554/eLife.47346.019

embryonic epithelium (*Figure 4C,F*). This cell cluster became visible in this location during embryo condensation, precisely when PGCs move beneath the epithelium in wildtype embryos (white asterisk in *Figure 2G,H*). Thus, in the absence of Fog signaling and primitive pit formation, the putative germ cells become mislocalized to the apical side of the embryonic epithelium (*Figure 3*).

## Fog signaling is involved in, but not required for, mesoderm internalization

We next asked whether Fog signaling plays a role in mesoderm internalization in *Tribolium*. As described earlier, *Drosophila* Fog signaling is required for the formation of a deep ventral furrow, but mesodermal cells still internalize in Fog signaling mutant embryos (*Seher et al., 2007*; *Sweeton et al., 1991*; *Zusman and Wieschaus, 1985*).

In *Tribolium*, mesoderm internalization occurs at the ventral side of the embryo like in *Drosophila*, but the mode of internalization is less uniform (*Handel et al., 2005*). Shortly after primitive pit formation, mesodermal cells flatten and constrict apically (*Handel et al., 2005*), causing the formation of a ventral furrow that is shallow at the anterior (*Figure 5A1 and A2*) and deeper at the posterior (*Figure 5A3*, *Figure 5—figure supplements 1* and *2A* for outlines of apically constricting cells). After serosal window closure, the mesoderm is fully internalized and the left and right ectodermal plates fuse along the ventral midline (*Figure 5B*).

Due to the dynamic nature of mesoderm internalization, it was important for us to compare carefully stage matched control and KD embryos. To do this, we carried out timed embryo collections and, in addition, examined the number of segments specified in these embryos (via analysis of *Tc-gsb* expression, *Davis et al., 2001*). At 19–21 hr after egg lay (AEL) (at 25˚C), control embryos had four trunk *Tc-gsb* stripes and had completely internalized their mesoderm (*Figure 5B and E*). *Tc-fog* KD embryos of the same age also had four trunk *Tc-gsb* stripes, and while some mesodermal cells exhibited apical constrictions, gastrulation was not complete (*Figure 5C and F*; *Figure 5—figure supplements 2* and *3B,D*). In anterior positions, gastrulation in KD embryos looked similar to 16–18 hr old control embryos (*Figure 5A1, C1*), while in middle and posterior regions, KD embryos showed a shallower furrow than that of control embryos at corresponding anterior-posterior (AP) positions (*Figure 5D,F*).

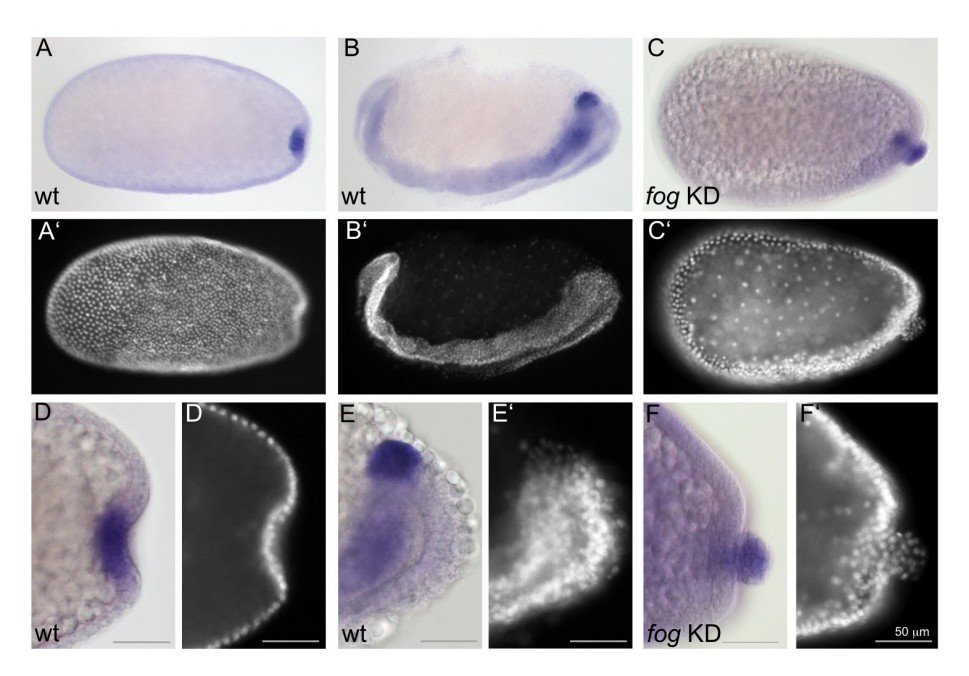

**Figure 4.** Fog signaling affects the positioning of the primordial germ cells. Whole mount ISH for the germ cell marker *Tc-tapas*. (A, B, D, E) Wildtype. (C, F) *Tc-fog* KD. (A–C) Optical sagittal sections of whole embryos. (D–F) Optical sagittal sections of posterior regions. (A'–F') DAPI staining of the respective embryos. (A, D) Wildtype embryo at primitive pit stage. (B, E) Wildtype embryo at early germ band extension stage. (C, F) *Tc-fog* KD embryos at stage corresponding to primitive pit stage in wildtype. Anterior is left, ventral is down.
DOI: https://doi.org/10.7554/eLife.47346.021

Despite the delay of mesoderm morphogenesis and the reduction in furrow depth in posterior positions, *Tc-fog* KD embryos eventually internalized the mesoderm. As in control embryos undergoing germ band extension, KD embryos possessed segmental clusters of *Tc-twi* expressing cells localized on the basal side of the ectoderm (*Figure 5G,H*; *Figure 5—figure supplement 4*). This situation is similar to *Drosophila*, where loss of *fog* affects coordination and speed of ventral furrow formation, but does not prevent mesoderm internalization.

### Regulation of *Tc-fog* and *Tc-mist* expression

We next investigated how Fog signaling is regulated in *Tribolium*. Like in *Drosophila*, ventral tissue specification in *Tribolium* depends on Toll signaling; *Tc-Toll* KD leads to a complete loss of mesoderm and ventral ectoderm fates (*Moussian and Roth, 2005*; *Nunes da Fonseca et al., 2008*; *Roth et al., 1989*). Therefore, we reasoned that the ventral stripe of *Tc-fog* expression is likely also dependent on Toll signaling. Indeed, pRNAi for *Tc-Toll* resulted in the loss of ventral *Tc-fog* expression (*Figure 6B*). *Tc-mist* expression, on the other hand, remained in the primitive pit region (*Figure 6I*). The serosa expression of each gene was also affected by *Tc-Toll* KD; *Tc-fog* became expressed in an expanded, DV symmetric domain, while *Tc-mist* showed weak uniform expression (*Figure 6B,I*). These changes reflect the expansion and dorsalization of the serosa upon *Tc-Toll* KD (*Nunes da Fonseca et al., 2008*).

To further dissect the ventral regulation of *Tc-fog* and *Tc-mist*, we analyzed the patterning genes downstream of Toll signaling. In both *Drosophila* and *Tribolium*, the transcription factors *twi* and *snail* (*sna*) are co-expressed in a ventral stripe (*Leptin and Grunewald, 1990*; *Sommer and Tautz, 1994*; *Stappert et al., 2016*). In *Drosophila*, both genes are required together to activate mesodermal *fog* expression (*Costa et al., 1994*), while *sna* alone is largely sufficient to activate mesodermal *mist* expression (*Manning et al., 2013*). In *Tribolium* we found that *Tc-mist* expression was unchanged following *Tc-twi* pRNAi (as expected based on their non-overlapping domains of expression) (*Figure 6J*), but *Tc-fog* was also unaffected (*Figure 6C*). Mesodermal *Tc-sna* expression is

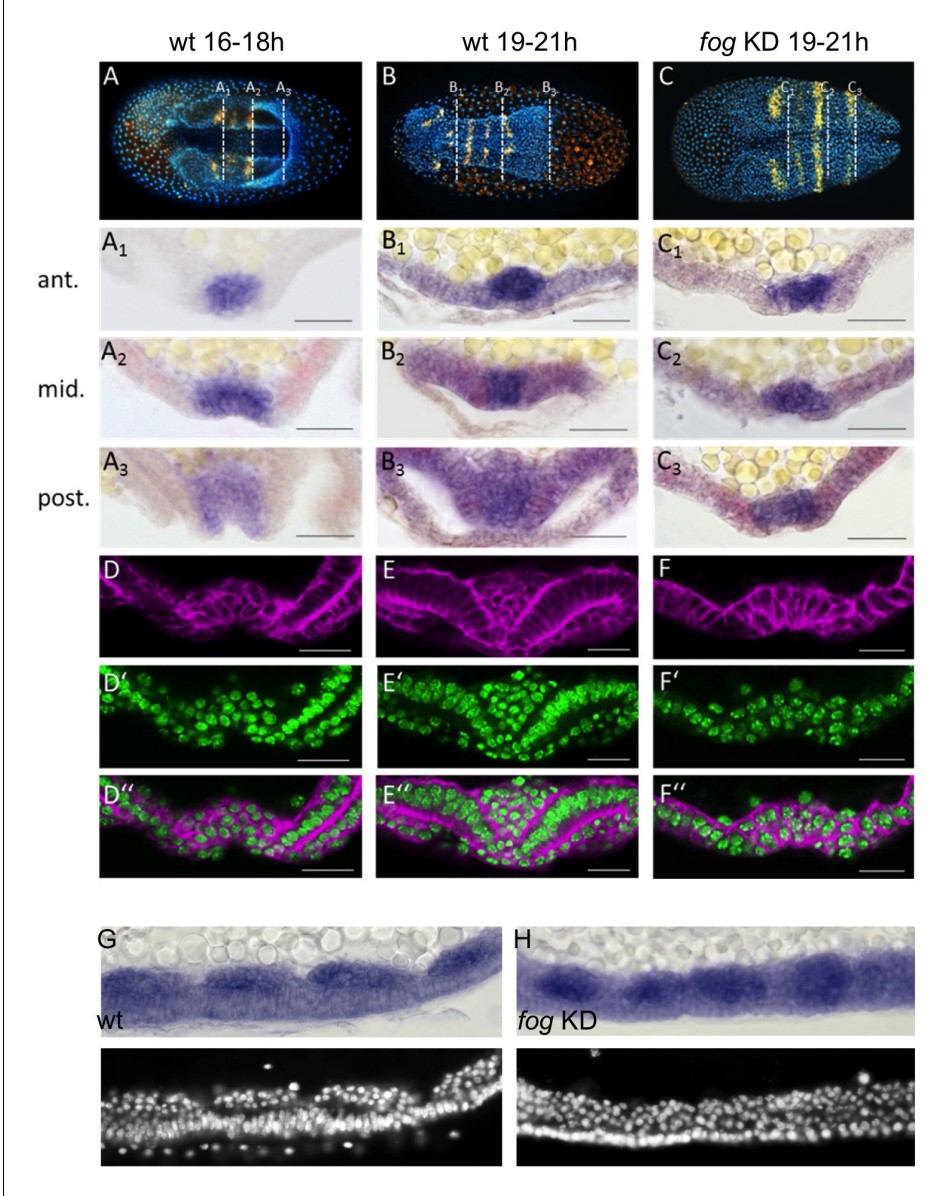

**Figure 5.** *Tc-fog* RNAi delays mesoderm internalization. (A–C) Ventral views of whole mount embryos (anterior left) stained for the segmental marker *Tc-gsb* (yellow), nuclei (DAPI; blue). Embryos are also stained for *Tc-twi* expression but this is only visible in A (dark blue ventral domain). (A–B) Wildtype at horseshoe or early germband extension stage with two or four trunk *Tc-gsb* stripes, respectively. (C) *Tc-fog* KD at a stage corresponding to (B) with four trunk *Tc-gsb* stripes (the forth stripe is visible in lateral views). (A₁–C₃) transverse cryosections of the embryos shown in (A–C) with *Tc-twi* expression (blue) and *Tc-gsb* expression (red). The position of each section is indicated by a white dashed line in (A–C). (D–F'') Transverse cryosections at posterior positions of embryos corresponding in age to those in (A–C) showing F-actin (phalloidin; magenta) and nuclei (sytox; green). (G, H) Sagittal cryosections of embryos during germband elongation (anterior left) showing *Tc-twi* expression (dark blue). (G', H') nuclear (DAPI) staining of respective embryos. Only a portion of the germband comprising four segments is shown. For corresponding sections showing the entire embryo see *Figure 5—figure supplement 4*. The different appearance of the mesoderm upon *Tc-fog* KD compared to wildtype is likely due to the delayed maturation of the mesoderm and the overall aberrant morphogenetic movements of KD embryos. For all embryos the serosa was removed during preparation. The cell sheet covering the ventral side of control embryos is the amnion (am) or amnion/dorsal ectoderm (am/de). Staging was done at 25°C.

DOI: https://doi.org/10.7554/eLife.47346.022

The following figure supplements are available for figure 5:

*Figure 5 continued on next page*

*Figure 5 continued*

**Figure supplement 1.** Mesoderm internalization varies along the AP axis in *Tribolium*.
DOI: https://doi.org/10.7554/eLife.47346.023
**Figure supplement 2.** Apical constrictions during mesoderm internalization.
DOI: https://doi.org/10.7554/eLife.47346.024
**Figure supplement 3.** Delayed mesoderm internalization upon *Tc-fog* KD.
DOI: https://doi.org/10.7554/eLife.47346.025
**Figure supplement 4.** The mesoderm is internalized upon *Tc-fog* KD.
DOI: https://doi.org/10.7554/eLife.47346.026

completely dependent on *Tc-twi* (*von Levetzow, 2008*), and, therefore, *Tc-fog* expression is not regulated by either *Tc-twi* or *Tc-sna*. Instead, ventral *Tc-fog* expression must depend on other zygotic factors in *Tribolium*.

To analyze the influence of the AP patterning system on *Tc-fog* and *Tc-mist* expression, we performed pRNAi for *Tc-caudal* (*Tc-cad*) (*Copf et al., 2004*; *Schoppmeier et al., 2009*) and *Tc-torso-like* (*Tc-tsl*) (*Schoppmeier and Schröder, 2005*). In *Drosophila*, *caudal* is required for the posterior, but not ventral, domain of *fog* expression (*Wu and Lengyel, 1998*). In contrast, KD of *Tc-cad* resulted in the loss of ventral *Tc-fog* expression and the appearance of a new domain of expression in the posterior-dorsal region of the embryo (asterisk in *Figure 6D*). Thus, *Tc-cad* both activates *Tc-fog* expression within the mesoderm and inhibits *Tc-fog* expression at the posterior of the embryo. Expression of *Tc-mist* was not notably altered after *Tc-cad* KD (*Figure 6K*).

*Tc-tsl* is a component of the terminal patterning system that specifies the anterior and posterior extremities of the AP axis (*Schoppmeier and Schröder, 2005*; *Schroder et al., 2000*). KD of *Tc-tsl* did not significantly affect mesodermal *Tc-fog* expression, but posterior *Tc-mist* expression was abolished (*Figure 6E and L*). This result matches published descriptions of *Tc-tsl* KD, which describe loss of normal posterior folding (*Schoppmeier and Schröder, 2005*).

Taken together, our results show that ventral expression of *Tc-fog* requires a combination of DV patterning (*Tc-Toll*) and AP patterning (*Tc-cad*) inputs, while the posterior expression of *Tc-mist* is controlled by the terminal patterning system (*Tc-tsl*) (*Figure 6—figure supplement 1*).

## The role of local *Tc-fog* expression for posterior folding

In *Drosophila*, the timing and location of *fog* expression is tightly linked with its function (*Bailles et al., 2019*; *Costa et al., 1994*; *Lim et al., 2017*). In contrast, *Tc-fog* is highly expressed in the serosa and posterior mesoderm, while its expression is conspicuously absent (or not detectable) from the posterior of the embryo where it is most prominently required, for the initiation of primitive pit and posterior amniotic fold formation. To approach this problem, we analyzed how each *Tc-fog* expression domain contributes to primitive pit and posterior amniotic fold formation. Specifically, we removed each *Tc-fog* domain individually or simultaneously and monitored the impact on the morphogenetic movements of the respective embryos.

To start, we deleted the serosal domain of *Tc-fog* (without affecting the ventral domain; *Figure 7—figure supplement 1*) by knocking down *Tc-zen1* to prevent serosa specification (*van der Zee et al., 2005*). This treatment had no detectable impact on primitive pit indentation or on the initiation of the posterior amniotic fold (*Figure 7—figure supplement 1*). During further development, posterior amniotic fold formation fails to achieve full encapsulation of the embryo, but this is likely due to the lack of a serosa in *Tc-zen1* KD embryos (*Panfilio et al., 2013*; *van der Zee et al., 2005*). However, the presence of a primitive pit and early posterior amniotic fold suggest that serosal *Tc-fog* expression is not essential for early morphogenetic events at the posterior of the embryo. Therefore, the ventral expression of *Tc-fog* might provide the source for Fog ligand required for posterior folding.

To test this assumption, we knocked down *Tc-Toll* to delete the ventral *Tc-fog* expression domain while maintaining high levels of *Tc-fog* expression in the serosa. Such embryos nonetheless still form a primitive pit-like indentation, and then a deep rotationally symmetric invagination at the posterior pole (*Figure 7A,B*). The distinctive appearance of the posterior folded tissue in *Tc-Toll* KD embryos could represent a secondary morphological consequence of germ rudiment condensation in a rotationally symmetric embryo. To test whether these tissue folds are truly Fog-dependent, we

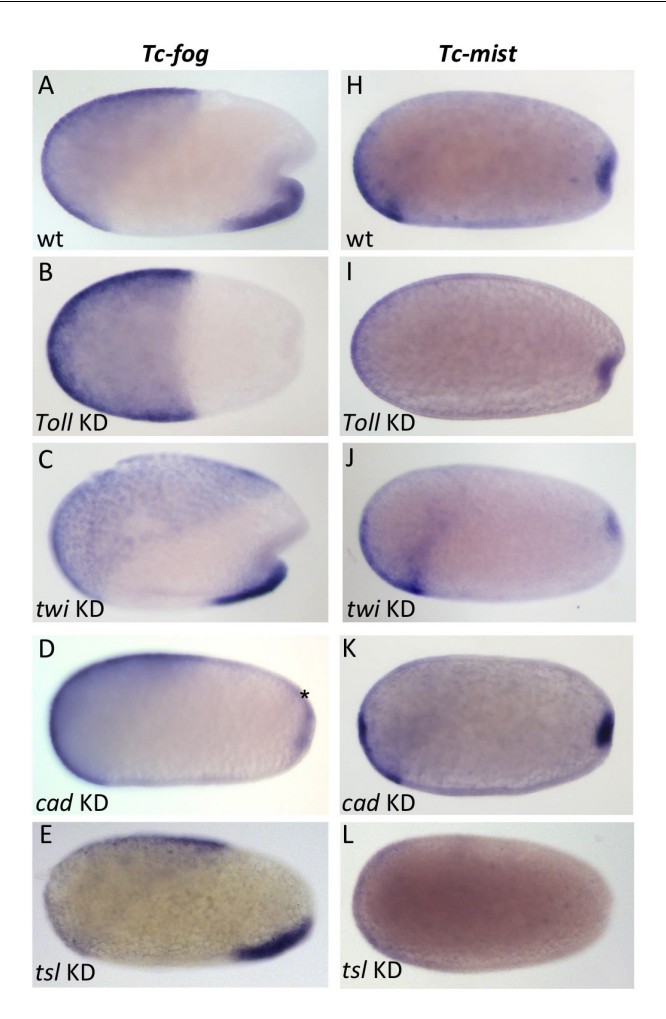

**Figure 6.** Regulation of *Tc-fog* and *Tc-mist* expression. Whole mount ISH for *Tc-fog* (**A–E**) and *Tc-mist* (**H–L**) expression in wildtype embryos (**A, H**) and embryos in which DV and AP genes have been knocked down (B-E, I-L; specific KD shown in panels). All embryos are at primitive pit stage except A and C, which are at the early posterior amniotic fold stage. The asterisk in (**D**) indicates the appearance of weak *Tc-fog* expression within a posterior-dorsal domain.

DOI: https://doi.org/10.7554/eLife.47346.027

The following figure supplement is available for figure 6:

**Figure supplement 1.** Regulation of *Tc-fog* and *Tc-mist* by DV and AP patterning genes.

DOI: https://doi.org/10.7554/eLife.47346.028

simultaneously knocked down *Tc-Toll* and *Tc-fog*. In nearly all double-KD embryos (87%, N = 81), posterior folding was indeed abolished (*Figure 7C*, *Video 8*). Thus, Toll KD embryos possess morphogenetic movements corresponding to the primitive pit and posterior amniotic fold of control embryos although they lack detectable *Tc-fog* expression in the germ rudiment.

As Tc-Fog is an extracellular ligand, it is possible in the *Tc-Toll* KD embryos that Tc-Fog protein diffuses from the serosal domain to the posterior of the embryo to activate Tc-Mist and initiate the posterior morphogenetic events. To test this hypothesis, we knocked down both *Tc-Toll* and *Tc-zen1* to remove both domains of *Tc-fog* expression simultaneously. In such embryos, no *Tc-fog* expression was visible by RNA ISH (*Figure 7—figure supplement 2*). Nevertheless, these embryos formed a symmetric posterior invagination as in *Tc-Toll* single KD embryos (*Figure 7D*). Therefore, diffusion of Tc-Fog from the serosa domain to the posterior also does not account for posterior folding.

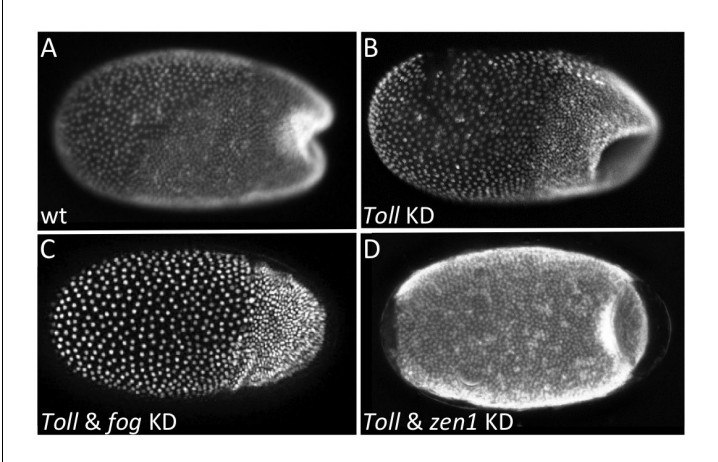

**Figure 7.** Local *Tc-fog* expression and posterior folding. Nuclear (DAPI) staining of wildtype (**A**) and KD (**B, C, D**; specific KD shown in panels) embryos at the early posterior amniotic fold stage. Anterior is left, ventral is down (where possible to discern).

DOI: https://doi.org/10.7554/eLife.47346.029

The following figure supplements are available for figure 7:

**Figure supplement 1.** *Tc-fog* expression after *Tc-zen1* KD.
DOI: https://doi.org/10.7554/eLife.47346.030
**Figure supplement 2.** *Tc-fog* expression after *Tc-Toll* and *Tc-zen1* KD.
DOI: https://doi.org/10.7554/eLife.47346.031

Two possibilities exist to explain this set of results. First, the primitive pit region of *Tc-Toll* KD embryos may harbor some *Tc-fog* transcript that, despite its low amounts, is sufficient to trigger the large-scale invagination of a symmetric posterior amniotic fold. Alternatively, there may be sufficient Tc-Fog protein remaining from the maternal expression of *Tc-fog* (*Dönitz et al., 2018*) to activate the Fog signaling pathway at the posterior of the embryo. However, both scenarios suggest that small amounts of Tc-Fog are sufficient to trigger large-scale folding specifically at the posterior pole.

## A novel role for Fog signaling in serosal spreading

Our finding that *Tribolium* Fog signaling is involved in mesoderm internalization and posterior amniotic fold formation fits with the classic function of this pathway in apical cell constriction. However, *Tc-fog* and *Tc-mist* are also expressed in the serosa, and what function (if any) these genes may have here is unknown. As described earlier, serosal cells become squamous as they spread to encapsulate the germband and yolk, but they also undergo intercalation as occurs during the analogous process of epiboly in zebrafish (*Benton, 2014*; *Köppen et al., 2006*). As Fog signaling has been implicated in cell intercalation in the *Drosophila* germband (*Kerridge et al., 2016*), we asked whether *Tribolium* Fog signaling may have the same function in the serosa. We first describe the wildtype pattern of intercalation before testing for a role of Fog signaling in this process.

We observed extensive cell intercalation during serosal spreading (*Figure 8*, *Videos 9–11*), but the pattern of intercalation varied across the embryo. During the first half of germband formation (stages 1–3), the serosa/germ rudiment boundary increases in length as it moves over the posterior pole. During this period, intercalation occurred at the posterior half of the

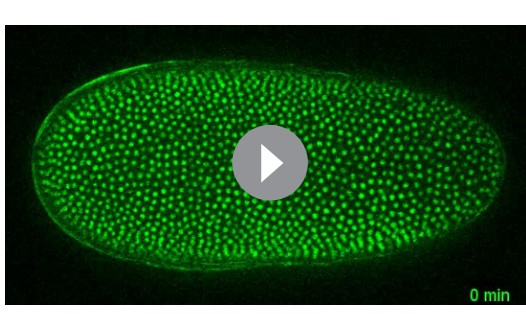

**Video 8.** Fluorescent live imaging of *Tc-Toll1* and *Tc-fog* double RNAi nGFP transgenic embryo. Maximum intensity projection of one egg hemisphere is shown with anterior to the left.
DOI: https://doi.org/10.7554/eLife.47346.032

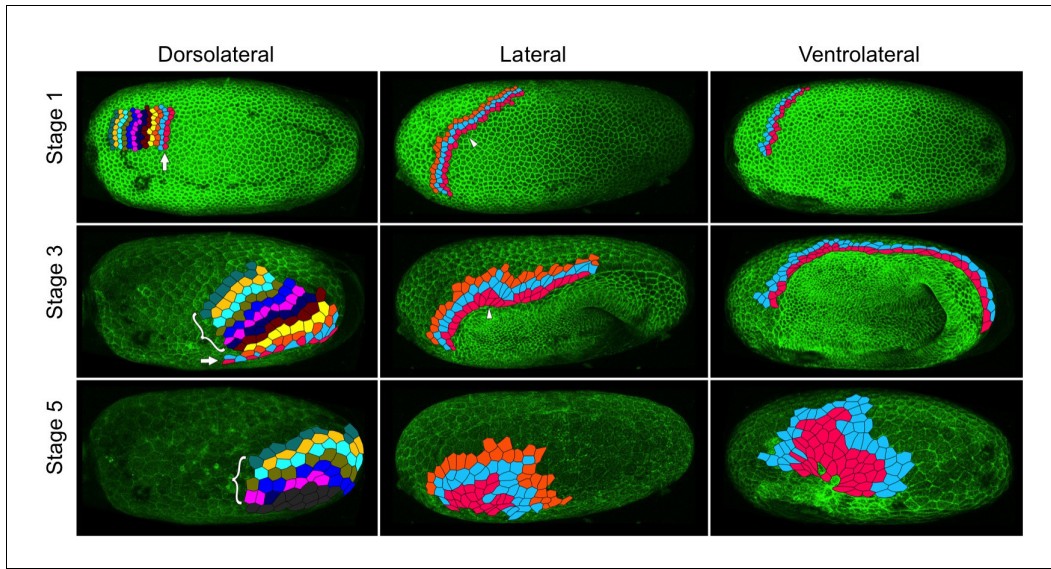

**Figure 8.** Serosal cells undergo intercalation during serosal expansion. Stills from confocal live imaging of wildtype embryos with cell membranes marked via transient expression of GAP43-YFP. The tracked cells are colored as rows parallel to the serosa/germ rudiment boundary (pink closest to the boundary). The arrow indicates the region where cells intercalate to increase the number of cells at the serosa/germ rudiment boundary during stages 1–3. The arrowhead indicates one example region where cells intercalate to leave the serosa/germ rudiment boundary during stages 1–3. The bracket indicates cells located away from the serosa/germ rudiment boundary that undergo intercalation during stages 1–5. The dark gray cells in the lower left panel went out of the frame of view and could not be tracked for the full movie. In the right panels, new cells were tracked from when they became visible halfway through embryo formation (middle panel). Cell outlines were manually drawn using projection views of individual timepoints and individual z-sections. All panels show maximum intensity projections of one egg hemisphere. Anterior is left, ventral is down.

DOI: https://doi.org/10.7554/eLife.47346.033

boundary to increase the number of cells at the boundary (*Figure 8* arrow in left panels, *Video 9*). In contrast, intercalation at more anterior regions caused cells to leave the boundary during the same period (*Figure 8* pink cells in middle and right panels, *Videos 10–11*). During the second half of germband formation (stages 4–5), serosal window closure causes the serosa/germ rudiment

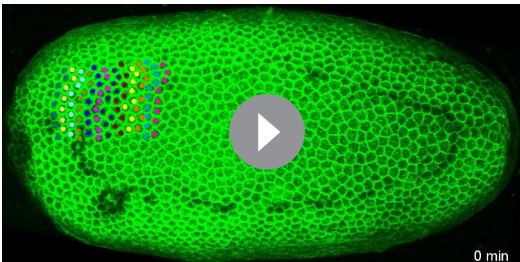

**Video 9.** Fluorescent live imaging of the dorsolateral side of a wildtype embryo transiently expressing GAP43-YFP. Serosa cells at/near the serosa/germ rudiment boundary were tracked and colored as rows (pink cells closest to the boundary). Only cells that were visible from the beginning of the timelapse are shown. Maximum intensity projection of one egg hemisphere is shown with anterior to the left and ventral to the bottom.

DOI: https://doi.org/10.7554/eLife.47346.034

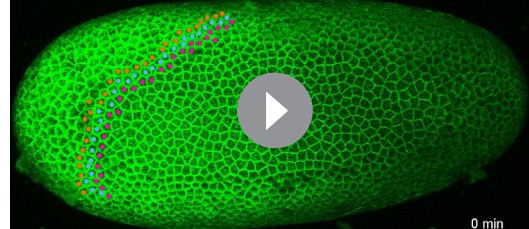

**Video 10.** Fluorescent live imaging of the lateral side of a wildtype embryo transiently expressing GAP43-YFP. Serosa cells at/near the serosa/germ rudiment boundary were tracked and colored as rows (pink cells closest to the boundary). Only cells that were visible from the beginning of the timelapse are shown. Maximum intensity projection of one egg hemisphere is shown with anterior to the left and ventral to the bottom.

DOI: https://doi.org/10.7554/eLife.47346.035

boundary to shrink to nothing. During this period, cells left the boundary all along its length (*Figure 8* pink cells in middle and right panels, *Videos 9–11*). Cell intercalation also occurred further away from the boundary during all stages of spreading (*Figure 8* bracketed cells in left panel, orange and pale blue cells in middle panel, and pale blue cells in right panel), and we observed the formation of multicellular rosettes (*Video 12*).

To test whether Fog signaling is involved in serosal cell intercalation, we used embryonic RNAi (eRNAi) to partially knock down *Tc-fog* to a level where phenotypic effects were visible but posterior amniotic fold formation and serosal expansion still occurred. We found that microinjection of *Tc-fog* dsRNA at 1 µg/µL caused roughly one-third of KD embryos (10 out of 29 embryos) to form a posterior amniotic fold and undergo serosal spreading. Embryos were co-injected with mRNA encoding the membrane marker GAP43-YFP to allow detailed live imaging and cell tracking (*Benton et al., 2013*). Cell intercalation was quantified by tracking roughly 70–100 cells at or near the serosa/germ rudiment boundary throughout serosa expansion and counting the number of intercalation events between tracked cells in four stage- and orientation-matched embryos (*Video 13*). Control embryos (injected with GAP43-YFP mRNA alone) showed 2.14 intercalation events per cell (171 cells tracked in two embryos) while *Tc-fog* partial KD embryos showed 2.56 intercalation events per cell (163 cells tracked in two embryos). This analysis shows that rather than decreasing intercalation in the serosa, reduction of Fog signaling led to a measurable increase in the relative amount of intercalation during serosal spreading.

A second possible role for Fog signaling in the serosa is an involvement in the cuboidal-to-squamous cell shape change that occurs during serosal spreading (*Benton et al., 2013*; *Handel et al., 2000*). To test this possibility, we quantified the extent of serosal cell spreading in *Tc-fog* partial KD embryos (like those described above). In such embryos, serosal cells still became squamous but differences in the extent of flattening existed at anterior and posterior positions (*Video 14*). The highly attenuated nature of these cells prevented quantification of cell height/volume, so we measured apical cell area after the completion of cell spreading. In control embryos (injected with GAP43-YFP mRNA alone), serosal cell area was variable, but this variability was evenly distributed along the AP axis ($\sigma$ = 145.96 µm$^2$; n = 535 cells across eight embryos; *Figure 9A,C*). In *Tc-fog* partial KD embryos, the variability in final serosal cell area was significantly increased compared with control embryos (p<0.001 Fisher's *F* test, $\sigma$ = 215.9 µm$^2$; n = 578 cells across seven embryos; *Figure 9B,D*). In addition, the serosal cells that covered the posterior half of the egg had larger surface areas than those in comparable positions in control embryos, while serosal cells in anterior regions showed the opposite pattern (*Figure 9B,D*). Based on our findings, we conclude that *Tribolium* Fog signaling has a novel role in serosal cells to coordinate the cuboidal-to-squamous cell shape transition that results in a uniformly thin layer of serosal cells surrounding the entire yolk and embryo.

## T48 enhances Fog signaling in *Tribolium*

In *Drosophila*, *fog* and *T48* both function during ventral furrow formation (*Kölsch et al., 2007*). We identified a single homolog of *T48* in *Tribolium* (hereafter referred to as *Tc-T48*), and while we could not detect localized *Tc-T48* expression by ISH, RNA-sequencing data suggested it is weakly expressed in embryos (*Dönitz et al., 2018*). Therefore, we tested whether *Tc-T48* has an embryo-wide enhancement function on Fog signaling in *Tribolium*.

To test for such a *Tc-T48* function, we microinjected embryos with *Tc-fog* dsRNA at 1 µg/µL (to partially KD *Tc-fog*) together with *Tc-T48* dsRNA. As described above, roughly a third of embryos microinjected with *Tc-fog* double-stranded RNA (dsRNA) (at 1 µg/µL) alone still formed a posterior amniotic fold and underwent serosa spreading (*Video 15*). When embryos were injected with both *Tc-fog* dsRNA and *Tc-T48* dsRNA (either by co-injection or sequential injections; n = 20 and 10, respectively) all embryos failed to form a posterior amniotic fold (*Video 15*). However, the double KD embryos did not show a more severe mesoderm internalization phenotype than the *Tc-fog* single KD embryos. Microinjection of *Tc-T48* dsRNA alone had no detectable effect on development (n = 10).

The statistically significant difference (p<0.001, Chi-Square test) in the phenotype caused by *Tc-fog* eRNAi alone versus *Tc-fog* and *TcT48* double RNAi, shows that *Tc-T48* has a morphogenetic function in *Tribolium*. Given the apparent lack of localized *Tc-T48* expression, we suggest that low levels of uniform expression play an embryo-wide role in enhancing Fog signaling.

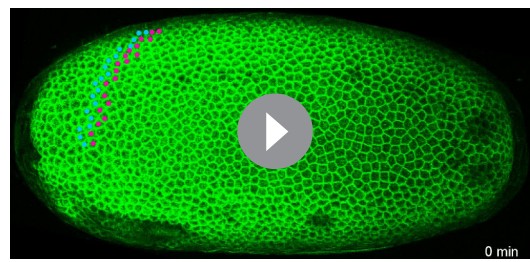

**Video 11.** Fluorescent live imaging of the ventrolateral side of a wildtype embryo transiently expressing GAP43-YFP. Serosa cells at/near the serosa/germ rudiment boundary were tracked and colored as rows (pink cells closest to the boundary). Roughly half the cells (at the anterior) are tracked from the beginning of the timelapse, while the rest are tracked from halfway through embryo formation (at 135 min) when they became visible. Maximum intensity projection of one egg hemisphere is shown with anterior to the left and ventral to the bottom.
DOI: https://doi.org/10.7554/eLife.47346.036

## Fog signaling is required for *Tribolium* blastoderm formation

In the course of analyzing the role of Fog signaling by embryonic RNAi, we injected different concentrations of dsRNA to vary the KD strength. While we recovered the phenotypes observed by parental RNAi with embryonic injections of 1 µg/µL dsRNA, injection of 3 µg/µL dsRNA yielded a phenotype that we had not obtained from pRNAi.

In the majority of KD embryos, major blastoderm-wide defects occurred during or prior to embryo condensation: 70% in *Tc-cta* KD, 70% in *Tc-mist* KD, 40% in *Tc-fog* KD versus 10% in control (n = 20 for each condition). Defects were highly variable in each KD and ranged from gaps in the blastoderm that became greatly enlarged during condensation to complete disintegration of the blastoderm prior to or during condensation (*Figure 10B–D*; *Videos 16* and *17*). In addition to the visible morphological defects, there was also a statistically significant delay in the development of *Tc-cta* and *Tc-mist* KD embryos (as measured using division 13 as a temporal landmark; *Figure 10—figure supplement 1*). This delay was not observed in *Tc-fog* KD embryos, which also had the lowest proportion of embryos with other blastoderm defects.

These phenotypes are unlikely to be artifacts caused by embryo handling or microinjection as they were seen at different proportions upon KD of each of the genes and were never seen at such high rates in control injections. As such, components of the Fog signaling pathway must also function during the formation of the blastoderm cell layer in *Tribolium*.

## The blastoderm function of Fog signaling is widely conserved

After finding that Fog signaling has key morphogenetic functions during embryonic development in *Tribolium*, we asked whether such functions are

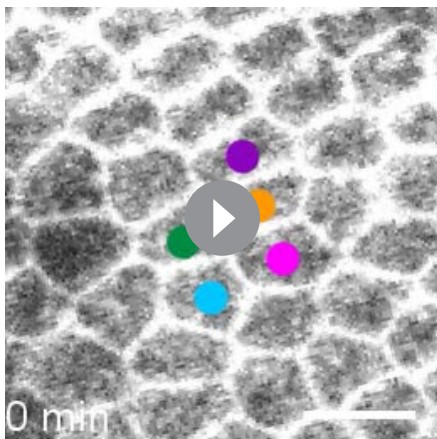

**Video 12.** Fluorescent live imaging of part of the forming serosa in a wildtype embryo transiently expressing GAP43-YFP. A group of cells intercalating via rosette formation are tracked. The field of view was manually stabilized to follow this group of cells. Anterior is to the left.
DOI: https://doi.org/10.7554/eLife.47346.037

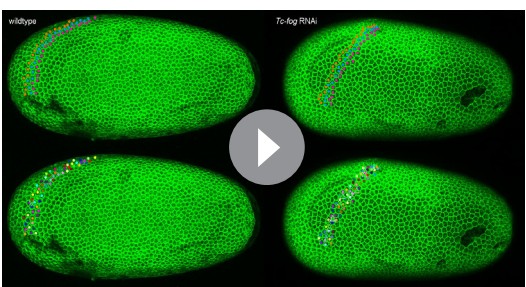

**Video 13.** Fluorescent live imaging of the wildtype and *Tc-fog* weak KD embryos transiently expressing GAP43-YFP. Serosa cells at/near the serosa/germ rudiment boundary were tracked. The top panels show cells coloured as rows (pink cells closest to the boundary). The bottom panels show cells colored randomly and intercalation events between tracked cells are shown with white triangles. Maximum intensity projection of one egg hemisphere is shown with anterior to the left and ventral to the bottom.
DOI: https://doi.org/10.7554/eLife.47346.038

widely conserved in insects. To answer this question, we functionally analyzed Fog signaling pathway components in two distantly related (*Misof et al., 2014*) hemimetabolous insects: the milkweed bug *Oncopeltus fasciatus* and the cricket *Gryllus bimaculatus.*

We identified single orthologs for *cta*, *mist* and *fog* in both species (hereafter called *Of-fog*, *Of-mist*, *Of-cta*, and *Gb-fog*, *Gb-mist*, *Gb-cta*, *Figure 1—figure supplement 2*). KD of each of these genes via pRNAi was performed in *Oncopeltus* and in *Gryllus* (except for *Gb-fog*, the KD of which resulted in adult lethality) and led to highly penetrant early phenotypes in both species. While control embryos (from parental injections of GFP dsRNA [*Oncopeltus*] or buffer [*Gryllus*]) formed a uniform blastoderm layer (n = 29 for *Oncopeltus*, n = 15 for *Gryllus*), each KD resulted in blastoderms that were interrupted by large holes along the entire AP axis (*Oncopeltus*: *Figure 10F–H*, 65% in *Of-fog* KD [n = 29], 64% in *Of-mist* KD [n = 25], 88% in *Of-cta* KD [n = 26]; *Gryllus*: *Figure 10J–L* and *Video 18*, 100% in *Gb-mist*, and *Gb-cta* KD [n = 19 each]).

While these early blastoderm defects prevented us from studying Fog function during later development in *Oncopeltus* and *Gryllus,* these phenotypes show that Fog signaling components have a deeply conserved requirement during blastoderm formation in insects.

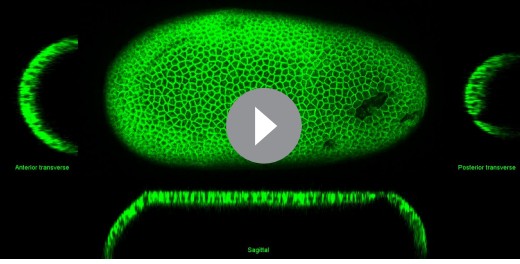

**Video 14.** Fluorescent live imaging of a *Tc-fog* weak KD embryo transiently expressing GAP43-YFP. Maximum intensity projection of one egg hemisphere with anterior to the left and ventral to the bottom is shown in the center. A transverse section near the anterior pole is to the left, a transverse section near the posterior pole is to the right, a sagittal section along the middle of the egg is shown to the bottom. DOI: https://doi.org/10.7554/eLife.47346.039

## Discussion

In this article we have shown that Fog signaling plays major morphogenetic roles during embryogenesis in the beetle *Tribolium* (*Figures 3* and *11*). Disruption of this pathway leads to severe defects during germband formation, including a complete loss of posterior amniotic fold formation, delayed mesoderm internalization, and mislocalization of PGCs. Fog signaling was also involved in the cuboidal-to-squamous cell shape change that occurs as the serosa spreads to cover the whole surface of the egg. Lastly, we found *Tribolium* Fog signaling to function during earlier stages of development: disruption of this pathway led to defects in the formation of a regular, continuous blastoderm epithelium. Functional analysis of Fog signaling in two distantly related insect species revealed this early blastoderm function to be widely conserved. In this discussion, we break down these diverse roles and discuss their importance for our understanding of the evolution of insect embryogenesis.

### Fog signaling has local morphogenetic functions during gastrulation

Fog signaling was discovered for its functions during early morphogenesis in *Drosophila* (*Zusman and Wieschaus, 1985*; *Sweeton et al., 1991*), but doubts were raised about whether this pathway functions during early embryogenesis in other insects (*Goltsev et al., 2007*; *Sweeton et al., 1991*; *Urbansky et al., 2016*; *Zusman and Wieschaus, 1985*). Here, we have shown that Fog signaling also functions during early development in a beetle, and that disruption of this pathway causes severe embryo-wide defects.

The most severe effect caused by disruption of Fog signaling in *Drosophila* is the loss of posterior gut fold formation (*Costa et al., 1994*; *Seher et al., 2007*; *Sweeton et al., 1991*). This posterior folding event in *Drosophila* not only internalizes the posterior endoderm (the posterior midgut proper), but also the hindgut (proctodeum), and it is required for the correct dorsal and anteriorwards extension of the germband. Therefore, it has been named amnioproctodeal invagination (*Campos-Ortega and Hartenstein, 1997*). *Tribolium* Fog signaling is also required for a folding event at the posterior of the blastoderm. However, since gut differentiation in *Tribolium* takes place only after the fully segmented germband has formed, it is not known how many cells involved in this folding event will later contribute to the gut (*Stanley and Grundmann, 1970*; *Berns et al., 2008*). Classical morphological descriptions as well as molecular work, though, suggest that posterior

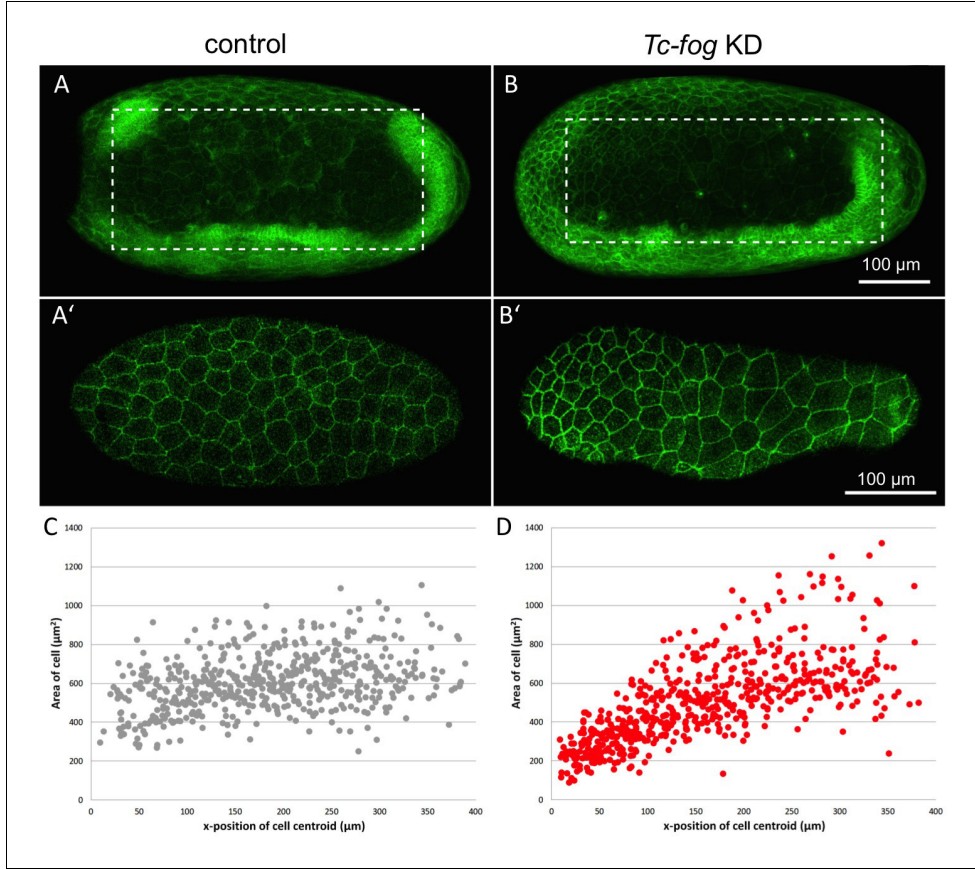

**Figure 9.** Fog signaling affects serosal expansion. (**A, B**) Stills from confocal live imaging of wildtype (**A**) and *Tc-fog* eRNAi (**B**) embryos with cell membranes marked via transient expression of GAP43-YFP. Embryos are undergoing germband elongation. (**A', B'**) Single optical section of the serosa from the dashed box region in (**A, B**) showing cells whose areas were measured. Quantification was performed on single optical section to avoid artefacts caused by curvature of the egg. (**C, D**) scatter plots showing serosa cell areas (wildtype (gray): 535 cells across eight embryos, *Tc-fog* eRNAi (red): 578 cells across seven embryos) together with AP position of cell centroids. Measurements were performed manually. (**A, B**) are average intensity projections of one egg hemisphere. Anterior is left, ventral is down.
DOI: https://doi.org/10.7554/eLife.47346.040

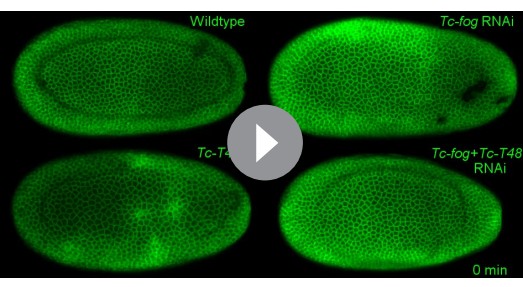

**Video 15.** Fluorescent live imaging of wildtype and *Tc-fog, Tc-T48*, and *Tc-fog* and *Tc-T48* double. RNAi embryos transiently expressing GAP43-YFP. Average intensity projections of one egg hemisphere are shown with anterior to the left and ventral to the bottom.
DOI: https://doi.org/10.7554/eLife.47346.041

midgut and proctodeum are derived from a region close to or encompassing the site of primitive pit formation (*Anderson, 1972b*; *Benton, 2018*; *Berns et al., 2008*; *Handel et al., 2000*; *Johannsen and Butt, 1941*; *Kispert et al., 1994*; *Stanley and Grundmann, 1970*; *Ullmann, 1964*). Indeed, one late phenotype of *Tribolium* embryos lacking Fog signaling concerns the hindgut, which points in the wrong direction, suggesting that *Tribolium* Fog is required for orienting the gut invagination towards the inside (*Figure 2—figure supplement 5*).

Although the role of the posterior amniotic fold for gut development in *Tribolium* needs further clarification, this fold has obvious consequences for early embryo topology: it causes the

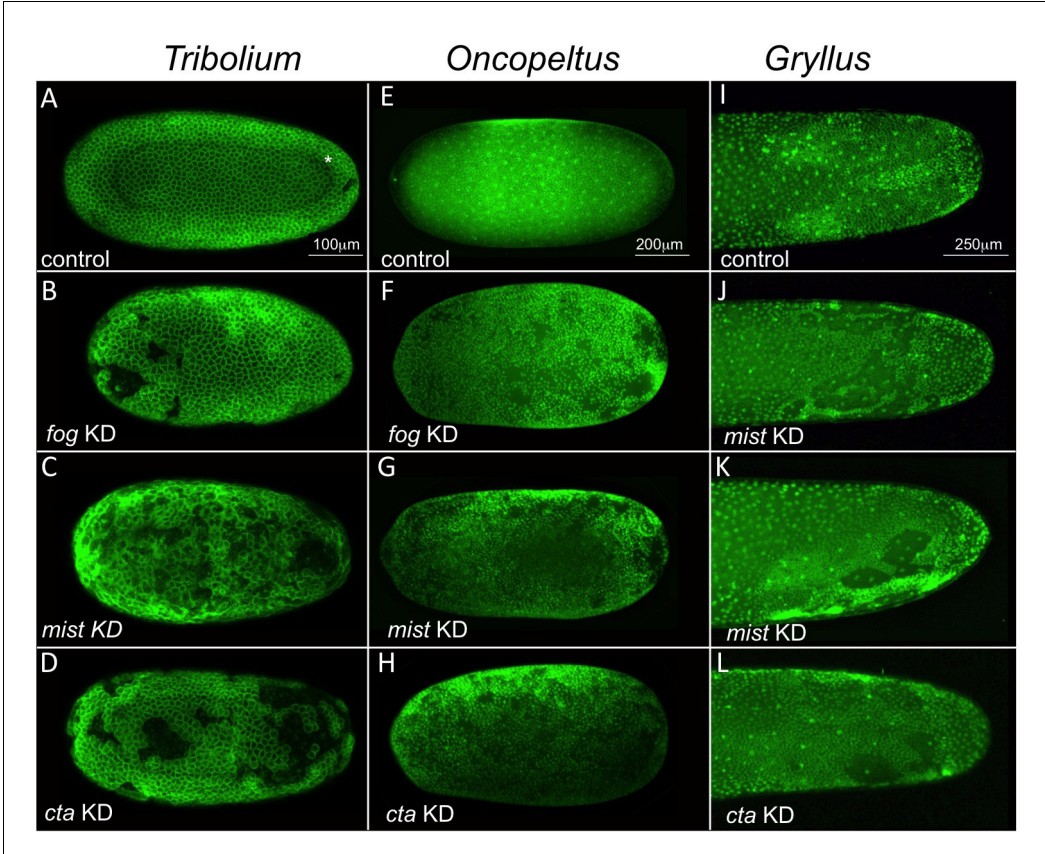

**Figure 10.** Fog signaling is required for blastoderm formation in *Tribolium*, *Oncopeltus* and *Gryllus*. (A–D) Stills from confocal live imaging of wildtype (A), and *Tc-fog, Tc-mist, Tc-cta* eRNAi (B–D) *Tribolium* embryos with cell membranes marked via transient expression of GAP43-YFP. Embryos are at late blastoderm stage. White asterisk in (A) indicated hole within the blastoderm which later closes. (E–H) *O. fasciatus* blastoderm stage wildtype (E), and *Of-fog, Of-mist, Of-cta* pRNAi (F–G) embryos with nuclei stained (Sytox) to reveal their distribution. (I–L) Stills from fluorescent live imaging of control (I), and *Gb-mist, Gb-cta* pRNAi (J–L) *G. bimaculatus* embryos with nuclei labeled via a *histone2B-eGFP* transgene. The phenotype of *Gb-mist* pRNAi is stronger in (J) than in (K). The latter embryo recovered during later development. (A–D) are average intensity projections of one egg hemisphere, (I–L) are maximum focus projections of one egg hemisphere. Anterior is left for all embryos. (I, J, L) are ventral views, (K) is a ventrolateral view with ventral down.

DOI: https://doi.org/10.7554/eLife.47346.042

The following figure supplement is available for figure 10:

**Figure supplement 1.** Developmental delay upon KD of Fog pathway components.

DOI: https://doi.org/10.7554/eLife.47346.043

amnion primordium/dorsal ectoderm to cover the ventral side of the embryo to form the amniotic cavity (*Handel et al., 2000*). Upon Fog signaling disruption and loss of the posterior amniotic fold, most of the germ rudiment tissue remains in an open configuration (*Figures 2* and *3*). This defective topology is reminiscent to that of wildtype embryos of *Drosophila* and other (cyclorrhaphan) dipteran species that do not become covered by an amnion-like tissue (Schmidt-Ott, 2000). As such, reduction/loss of early posterior Fog signaling may have contributed to evolution of the *Drosophila*-like mode of development. To address this question, more detailed descriptions of the genetic and morphogenetic events occurring during posterior development in other insect species are required.

The mechanisms of Fog signaling at the posterior pole in *Tribolium* pose an interesting riddle. While *Tc-mist* is expressed in the region where posterior folding is initiated, this region lacks detectable *Tc-fog* expression (*Figure 1*; *Figure 6A,H* and *Figure 6—figure supplement 1*). The area closest to the primitive pit that harbors *Tc-fog* expression is the ventrally abutting mesoderm (*Figure 1—figure supplement 4*). However, even if mesodermal *Tc-fog* expression is removed, like

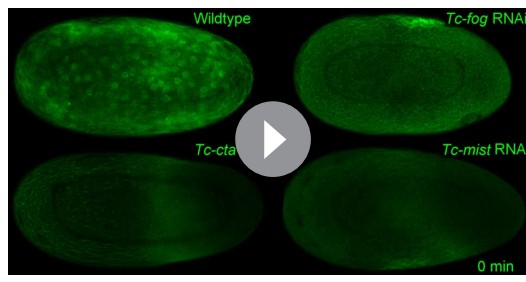

**Video 16.** Fluorescent live imaging of wildtype and *Tc-fog*, *Tc-cta*, and *Tc-mist* RNAi embryos transiently expressing GAP43-YFP. Average intensity projections of one egg hemisphere are shown with anterior to the left and ventral to the bottom (where possible to discern).
DOI: https://doi.org/10.7554/eLife.47346.044

in *Tc-Toll* KD embryos, posterior folding still takes place, suggesting that expression levels undetectable by our methods are sufficient to induce folding (*Figure 7*). This is in apparent contradiction to the extent of posterior folding, which results in a large-scale tissue re-arrangement. It is therefore possible that posterior folding in *Tribolium* involves some form of mechanical feedback amplification through which the folding event is less sensitive to the initial amount of Fog signaling, as was recently shown for *Drosophila* (*Bailles et al., 2019*).

Posterior folding in *Drosophila* is also driven by a cycle of cell deformations that require integrin-mediated cell adhesion to the inner eggshell (vitelline membrane) (*Bailles et al., 2019*). Such integrin-mediated anchoring was first described in *Tribolium*, where anterior anchoring exists during germband formation (*Münster et al., 2019*). The same integrin that is required for the anterior attachment point (*a*PS2, *Tc-inflated*) in *Tribolium* is also expressed at the posterior pole (*Münster et al., 2019*) suggesting a further similarity between *Drosophila* and *Tribolium* posterior folding events. However, any attachment at the posterior pole in *Tribolium* must be transient, as our live imaging did not reveal a static posterior attachment point as we previously observed at the anterior using the same approach (*Benton et al., 2013*).

The second major role of Fog signaling in *Drosophila* is during mesoderm infolding, and this function also appears to be conserved in *Tribolium*. Disruption of Fog signaling has very similar consequences for mesoderm internalization in *Tribolium* and *Drosophila*. In both cases, the mechanism and timing of mesoderm internalization is affected by loss of signaling, but the mesoderm is still able to internalize (*Seher et al., 2007*; *Zusman and Wieschaus, 1985*). The same is also true for the dipteran *Chironomus*, where disruption of Fog signaling has a measurable impact on mesoderm internalization but the pathway is not strictly required for the process (*Urbansky et al., 2016*).

Further evidence for conservation of Fog signaling function in mesoderm internalization comes from the cell shape changes caused by Fog signaling. While anterior regions of the *Tribolium* mesoderm do not express *Tc-fog* and internalize by forming only a shallow furrow, the posterior half of the mesoderm expresses *Tc-fog* and does form a deep furrow during internalization (*Figure 1M–P*; *Figure 5A* and *Figure 5—figure supplement 1*; *Figures 3* and *11*) (*Handel et al., 2005*). This result suggests that mesodermal

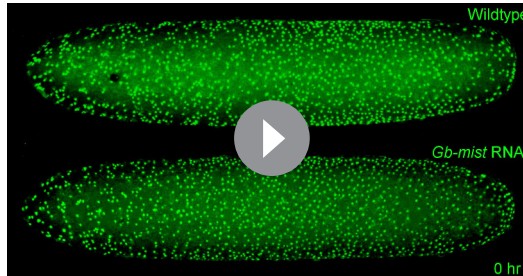

**Video 18.** Fluorescent live imaging of wildtype and *Gb-mist* RNAi histone2B-EGFP transgenic embryos. Maximum focus projections of one egg hemisphere are shown as ventral views with anterior to left.
DOI: https://doi.org/10.7554/eLife.47346.046

**Video 17.** Fluorescent live imaging of additional *Tc-fog*, *Tc-cta*, and *Tc-mist* RNAi embryos transiently expressing GAP43-YFP. Further examples of blastoderm formation defects to demonstrate the variability in the phenotypes. Average intensity projections of one egg hemisphere are shown with anterior to the left and ventral to the bottom (where possible to discern).
DOI: https://doi.org/10.7554/eLife.47346.045

cells that experience high levels of Fog signaling enhance furrow formation, while those that may experience lower signaling (i.e. via diffusion of Fog ligand from neighbouring cells) do not. This hypothesis is also supported by research on Fog signaling in *Chironomus* and *Drosophila*. In *Chironomus*, the mesoderm forms a shallow furrow during internalization, and while *fog* is expressed in this tissue, expression is notably weaker in the ventralmost part of the domain. Experimental over-activation of Fog signaling triggers the formation of a deep ventral furrow (*Urbansky et al., 2016*), suggesting a quantitative response to levels of Fog ligand. In *Drosophila*, quantitative analyses have shown that the accumulation of Fog ligand directly correlates with the degree of change in the cytoskeleton required for cell shape changes (*Lim et al., 2017*).

An interesting difference between *Drosophila* and *Tribolium* consists of the function of *T48*. While mesodermal expression of *T48* contributes to ventral furrow formation in *Drosophila* (*Kölsch et al., 2007*), *Tc-T48* lacks local expression. Double KD with *Tc-fog* did not block mesoderm internalization, but enhanced the posterior folding phenotype. Thus, instead of the local mesoderm-specific function of *T48* in *Drosophila* we suggest that *Tc-T48* has a weak embryo-wide function, which we could only detect at the posterior pole.

Taken together posterior folding and mesoderm internalization show different requirements for Fog signaling in *Tribolium*, as is also true in *Drosophila* (*Figure 11*). Posterior folding is absolutely dependent on Fog, although apparently low levels of signaling are sufficient to induce massive folding. On the other hand mesoderm internalization is not strictly dependent on Fog, which rather acts quantitatively to modulate the depth of the invagination furrow along the anterior-posterior axis.

## The role of Fog signaling in primordial germ cell positioning

One function for *Tribolium* Fog signaling that does not exist in *Drosophila* is the role in PGC positioning. We found that disruption of Fog signaling leads to *Tribolium* PGCs moving to the apical surface of the embryonic epithelium rather than being internalized basally (*Figures 3*, *4* and *11*). This new aberrant localization is comparable to the positioning of *Drosophila* PGCs (the pole cells) at the apical surface of the blastoderm (*Cinalli and Lehmann, 2013*).

We propose two possible scenarios for Fog's role in PGC internalization in *Tribolium*. One possibility is that PGC localization is due to a requirement for Fog signaling within the epithelial cells surrounding the PGCs. For instance, Fog-mediated apical constrictions of posterior blastoderm cells could bias the movement of PGCs to the basal side of the epithelium. When Fog signaling is disrupted, PGCs would carry out their normal developmental program and leave the epithelium, but the absence of apical constriction and primitive pit formation would cause the PGCs to localize to the apical side of the blastoderm. Alternatively, Fog signaling may directly control cell polarity within the PGCs, and it is the breakdown of this process that affects PGC localization. In addition, some combination of both sides of Fog activity may be true. Given that we observe a well-formed, albeit wrongly positioned, cluster of PGCs after loss of Fog signaling, it seems more likely that Fog signaling acts after PGC formation and without impairing cellular organization within the cluster.

Despite the lack of overt conservation of this PGC function, research in *Drosophila* does reveal a possibility as to how Fog signaling could be affecting *Tribolium* PGC development. In *Drosophila*, the GPCR Trapped in endoderm 1 (Tre1) is necessary in PGCs for their migration through the midgut epithelium (*LeBlanc and Lehmann, 2017*). Tre1 is activated by guidance cues and promotes germ cell migration by polarizing Rho1. In *Tribolium*, Fog signaling could potentially also act to polarize Rho1 via RhoGEF2 recruitment within PGCs and thereby effect their migration to the basal side of the epithelium.

## Fog signaling has tissue-wide functions in the blastoderm and serosa

The functions for *Tribolium* Fog signaling discussed above fit with the traditional role for this pathway in apical constriction. In contrast, the involvement of Fog signaling in serosal spreading and blastoderm formation are two processes that do not involve apical constriction (*Figure 11*).

During serosal spreading, Fog signaling acts in a process that is effectively the opposite of apical constriction: the expansion of the apical (and basal) cell surface to cause the cuboidal-to-squamous transition. To analyse this function, partial KD of Fog was most informative, as posterior folding and serosal spreading still occurred, but were no longer uniform throughout the tissue. Rather, cells closest to the dorsal serosa/germ rudiment border acquired greater apical surface areas, while the

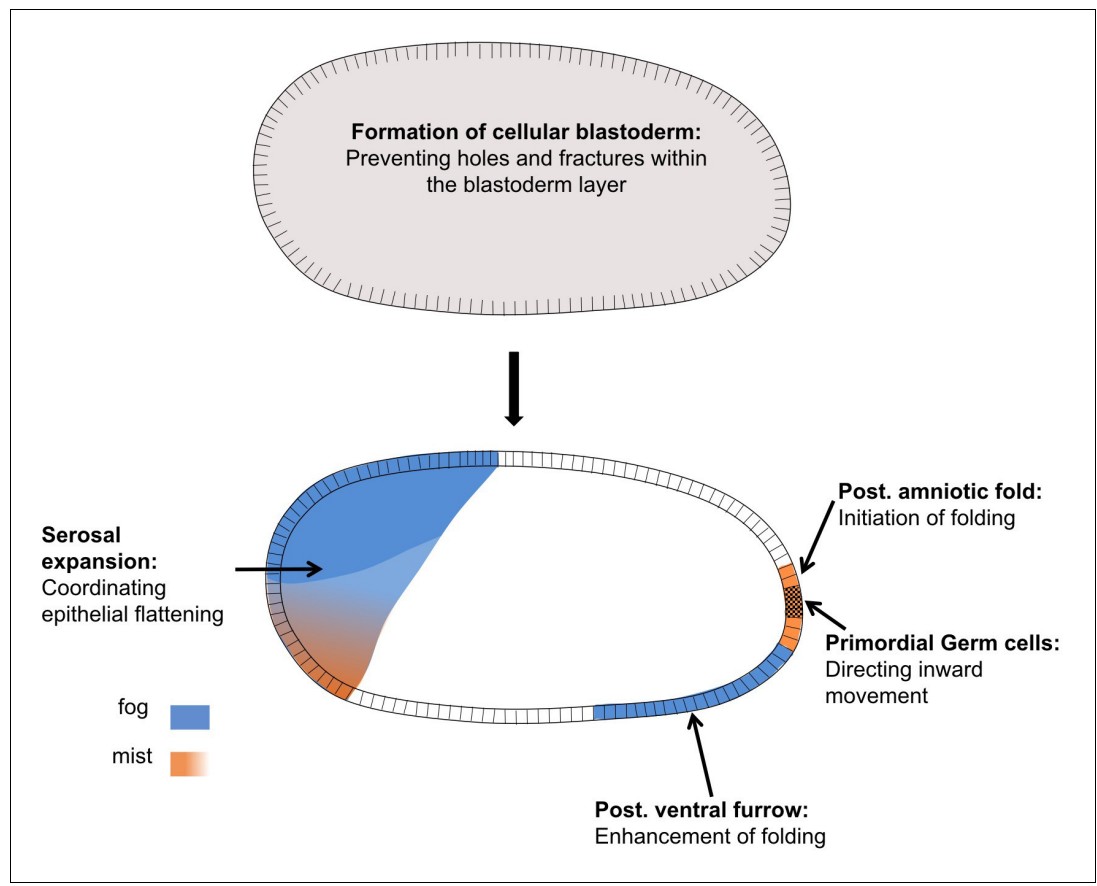

**Figure 11.** Distinct functions of Fog signaling in *Tribolium*. Summary schematic showing the different roles of fog signaling during early *Tribolium* embryogenesis.
DOI: https://doi.org/10.7554/eLife.47346.047

remaining serosa cells in fact had reduced surface areas compared to wildtype cells at corresponding positions (*Figure 9*).

Tissue forces have not been directly measured in *Tribolium*, but indirect evidence indicates that folding and condensation of the germ rudiment exert pulling forces on the serosa (*Münster et al., 2019*). If true, these forces would likely be greatest at the dorsal serosa/germ rudiment border. Despite this tissue-level anisotropy, spreading of serosal cells appears fairly uniform in wildtype embryos. This uniformity may in part be due to the serosal cell intercalation we describe (*Figure 8*). Such intercalation events could be triggered by local tension in the epithelium as described in other systems (e.g. *Aigouy et al., 2010*) and could thereby serve to maintain tissue fluidity and reduce anisotropy in tissue tension (*Tetley and Mao, 2018*; *Tetley et al., 2019*). A recent preprint examines this process in further detail (*Jain et al., 2019*).

We found that reduced Fog signaling caused an increase in the relative number of intercalation events and decreased uniformity in serosal cell spreading (*Figure 9*). The increase in intercalation could be due to increased anisotropy in epithelial tension caused by the defects in serosal cell spreading. Thus, the role of Fog signaling in the serosa could be to regulate the cuboidal-to-squamous cell shape transition to evenly spread the propagation of forces between neighbouring serosal cells. The use of a paracrine signaling pathway such as Fog signaling for this function makes sense, as it would allow tissue-wide coarse-graining via the extracellular distribution of ligand, buffering the degree to which cells experience different forces across the tissue. At the intracellular level, Fog signaling could be influencing the distribution of myosin to affect cell spreading. As the mechanisms underlying cuboidal-to-squamous cell shape transitions are not well understood (*Brigaud et al., 2015*; *Grammont, 2007*; *Wang and Riechmann, 2007*), and only descriptive (*Benton et al.,*

*2013*; *Panfilio and Roth, 2010*), but no mechanistic information on serosal spreading exists, substantial future work will be required to uncover the mechanism of Fog signaling during this important and widely conserved developmental event.

Fog signaling also appears to have a global tissue function during blastoderm formation in *Tribolium* (*Figure 11*) and two other species representing deep branches within the insects. In *Tribolium* we observed this phenotype only after embryonic injections of high amounts of dsRNA. Upon parental RNAi, which should affect both maternal and zygotic transcripts, cellularization was normal. However, parental KD also led to partial lethality and sterility of the injected individuals (e.g. for *Tc-mist* the survival rate was 77% (N = 200) and the surviving females produced a reduced number of eggs for only three days before becoming sterile). It is therefore likely that the eggs obtained following pRNAi represented an incomplete KD, despite the fact that the phenotypes we observed were highly penetrant (*Figure 2—figure supplement 4*). Embryonic RNAi circumvents the problem of adult lethality and sterility and apparently allows a more efficient KD of maternal and zygotic transcripts of Fog pathway components. In *Drosophila* problems with adult lethality and (certain types of) sterility can be overcome by producing mutant germ line clones. Using this technique *Drosophila* embryos were generated that completely lacked maternal and zygotic *fog* transcripts. However, these embryos did not show a stronger phenotype than embryos lacking only the zygotic transcripts (*Costa et al., 1994*; *Zusman and Wieschaus, 1985*). In particular, no defects during cellularization were observed.

The different requirement for Fog signaling during cellularization in *Drosophila* and three insects we have studied may be linked to variations in the modes of cellularization. The formation of a high-columnar blastoderm, which has been well studied in *Drosophila*, is an exception among insects and even among flies (*Bullock et al., 2004*; *van der Zee et al., 2015*). In many insects, cellularization generates a blastoderm of cuboidal cells, while some insects initially form individual cells that then migrate to form a continuous epithelium (*Ho et al., 1997*; *Nakamura et al., 2010*).

Little is known about the molecular mechanisms underlying the diverse modes of cellularization described above, but each mode will have its own mechanical requirements. For example, in *Drosophila* strong lateral adhesion between the highly columnar cells provides stability (*Mazumdar and Mazumdar, 2002*), while in *Tribolium* transitory holes and fractures between protocells are visible within the wildtype blastoderm (white asterix in *Figure 10*). Here, *Tribolium* Fog signaling may have a non-cell autonomous influence on actomyosin dynamics, increasing the stability and robustness of the epithelium to prevent fracture formation or heal existing fractures. However, the way in which this may occur at a mechanistic level is a completely open question. The discovery of special Innexin-based cell junctions that are essential for cellularization in *Tribolium,* but do not exist in *Drosophila,* highlights the potential diversity in mechanisms underlying cellularization in different insects (*van der Zee et al., 2015*). Thus, as with the spreading of the serosa, future work is required to uncover the mechanism of Fog signaling during blastoderm formation in *Tribolium* and other insects.

While the mechanism remains unknown, the requirement for Fog signaling during blastoderm formation in phylogenetically diverse insects suggests this to be a widely conserved function. Finally, while gastrulation is even more variable in non-insect arthropods than in insects, nearly all described arthropod species also form a cellular blastoderm that at least partially covers the surface of the egg (*Anderson, 1973*; *Wanninger, 2015a*; *Wanninger, 2015b*). As such, the early function of Fog signaling during blastoderm formation may be the most ancestral role of this pathway in arthropods.

## Materials and methods

### Strains

*Tribolium castaneum* strains: San Bernandino wildtype (*Brown et al., 2009*), nuclear GFP (nGFP) (*Sarrazin et al., 2012*), LifeAct-GFP (*van Drongelen et al., 2018*) were cultured as described (*Brown et al., 2009*). *Oncopeltus fasciatus* was cultured as described (*Ewen-Campen et al., 2011*). *Gryllus bimaculatus* wildtype strain (*Donoughe and Extavour, 2016*) and pXLBGact Histone2B:eGFP (*Nakamura et al., 2010*) was kept as described (*Donoughe and Extavour, 2016*).

### cDNA cloning

The primers used for in-situ hybridization and dsRNA synthesis were designed by using the new *T. castaneum* genome assembly (*Dönitz et al., 2018*). Primer design, RNA extraction and cDNA synthesis were carried out using standard protocols. For *Gryllus bimaculatus,* a new assembled transcriptome was used to design primers for *Gb-cta* and *Gb-mist*. The Advantage GC 2 PCR Kit (Takara) was used for gene cloning. All relevant genes, their corresponding Tc-identifiers and the primers used in this work are listed in the supplement (*Supplementary file 1*).

### dsRNA synthesis, parental and embryonic

RNAi dsRNA preparation, pupae and adult injections followed *Posnien et al. (2009)*. Embryonic RNAi was performed as described (*Benton et al., 2013*). For *Gryllus bimaculatus*, a 1184 bp fragment of *Gb-cta* and a 967 bp fragment of *Gb-mist* was used to knock down gene function. For both genes, the knockdown was performed in two independent experiments injecting 7 µg and 10 µg of dsRNA per animal, respectively. The dsRNA solution was injected into the proximal joint of the coxa of the second and third leg. For each experiment, four adult females of the pXLBGact Histone2B: eGFP line were injected and embryos of the second egg lay (collected about one week after injection) were analyzed via live imaging.

### In-situ hybridisation, immunohistochemistry

Single and double ISH were performed essentially as described (*Schinko et al., 2009*). For staining of cell membranes, Alexa Fluor 555/568 Phalloidin (Molecular Probes, life technologies) was used. Nuclear counterstaining was performed using DAPI (Invitrogen) or Sytox Green (Thermo Fisher) as previously described (*Nunes da Fonseca et al., 2008*).

### Cryosections

Embryos were embedded in a melted agarose/sucrose solution (2% agarose, 15% sucrose in PBS). After the agarose cooled down, blocks of agarose containing the embryos were cut and incubated in a solution of 15% sucrose in PBS overnight. The blocks were fixed to the specimen block using Tissue-Tek O.C.T. (Sakura). After shock freezing in −80°C isopentan, the blocks were transferred to a Cyrostat (Leica CM 1850) and sliced at −20°C (30µm-thick sections). The sections were mounted on Superfrost Ultra Plus microscope slides (Thermo Scientific) and dried over night at RT. Phalloidin and Sytox staining directly on the sections was performed following the standard protocols using a humidity chamber.

### Live imaging

Confocal time-lapse imaging of *Tribolium* embryos injected with GAP43-YFP mRNA was performed as previously described (*Benton et al., 2013*; *Benton, 2018*) at 25–32°C at time intervals from 2 to 10 min between timepoints using 20x, 40x, or 63x objectives. For live imaging of the posterior poles, eggs were propped up vertically (resting against another egg for stability) on a glass-bottomed Petri dish (MatTek) with their posterior against the glass. Live imaging transgenic nuclear GFP (nGFP) (*Sarrazin et al., 2012*) or LifeAct-GFP (*van Drongelen et al., 2018*) embryos was done at room temperature using the Zeiss AxioImager.Z2 in combination with an Apotome.2 and movable stage (Zen2 Blue).

For imaging *Gryllus bimaculatus* embryos we used a Zeiss AxioZoom.V16, equipped with a movable stage. *Gryllus* embryos were placed on 1.5% agarose and were covered with Voltalef H10S oil (Sigma). Imaging was performed at 25–27°C.

### Data analysis and software

Image analysis was performed in FIJI (*Schindelin et al., 2012*). Serosa cell areas were measured manually and cells at the periphery of the section were not included. Cell tracking and quantification of cell intercalation was performed manually on 4D hyperstacks and various projections (maximum/average intensity, sum slices) using mTrackJ (*Meijering et al., 2012*). Confocal data were rotated using TransformJ with quintic B-spline interpolation (*Meijering et al., 2001*). Additional plugins include Bioformats Importer (*Linkert et al., 2010*), Image Stabiliser (*Li, 2008*), StackReg (*Thevenaz et al., 1998*), and Bleach Correction (*Miura et al., 2014*).

Box/violin plots were generated using PlotsOfData (*Postma and Goedhart, 2019*). Figures and schematics were created using Powerpoint and the open source software Inkscape.

## Acknowledgements

We thank Nathan Kenny for assembling the *Gryllus bimaculatus* transcriptome, S Noji and T Mito for providing *Gryllus* strains. NF, KHC and SR were supported by DFG CRC 680, MAB by a Humboldt Fellowship for Postdoctoral Researchers and DFG Research Fellowship 407643416, RNdF and CvL by the International Graduate School in Genetics and Functional Genomics, Cologne University. Research in the R NdF lab is supported by the following Brazilian agencies: *CNPq, FAPERJ and CAPES.* MP was supported by a UoC Postdoc Grant, MSH by DFG FOR 1234, DS by a Ph.D. fellowship of the Boehringer Ingelheim Fonds, KAP by the Emmy Noether Program of the DFG (grant PA 2044/1-1), JAL was supported by grant R03 HD078578 from the US National Institutes of Health.

## Additional information

### Funding

| Funder | Grant reference number | Author |
|---|---|---|
| Deutsche Forschungsgemeinschaft | CRC 680 | Nadine Frey<br>Siegfried Roth |
| University of Cologne | International Graduate School in Genetics and Functional Genomics | Rodrigo Nunes da Fonseca<br>Cornelia von Levetzow |
| Conselho Nacional de Desenvolvimento Científico e Tecnológico | | Rodrigo Nunes da Fonseca |
| Coordenação de Aperfeiçoamento de Pessoal de Nível Superior | | Rodrigo Nunes da Fonseca |
| Deutsche Forschungsgemeinschaft | RU 1234 | Muhammad Salim Hakeemi |
| Boehringer Ingelheim Fonds | PhD fellowship | Dominik Stappert |
| Deutsche Forschungsgemeinschaft | Emmy Noether Program PA 2044/1-1) | Kristen A Panfilio |
| National Institutes of Health | R03 HD078578 | Jeremy A Lynch |
| Alexander von Humboldt Foundation | Postdoctoral Fellowship | Matthew Alan Benton |
| Deutsche Forschungsgemeinschaft | | Kai H Conrads |
| University of Cologne | Postdoctoral grant | Matthias Pechmann |
| Deutsche Forschungsgemeinschaft | DFG Research Fellowship 407643416 | Matthew Alan Benton |
| Fundação Carlos Chagas Filho de Amparo à Pesquisa do Estado do Rio de Janeiro | | Rodrigo Nunes da Fonseca |

The funders had no role in study design, data collection and interpretation, or the decision to submit the work for publication.

### Author contributions

Matthew Alan Benton, Conceptualization, Formal analysis, Investigation, Visualization, Methodology, Writing—review and editing; Nadine Frey, Formal analysis, Investigation, Methodology, Writing— review and editing; Rodrigo Nunes da Fonseca, Conceptualization, Formal analysis, Investigation, Methodology, Writing—review and editing; Cornelia von Levetzow, Conceptualization, Formal analysis, Investigation, Methodology; Dominik Stappert, Formal analysis, Investigation; Muhammad

Salim Hakeemi, Kai H Conrads, Investigation, Writing—review and editing; Matthias Pechmann, Investigation, Methodology, Writing—review and editing; Kristen A Panfilio, Formal analysis, Investigation, Writing—review and editing; Jeremy A Lynch, Resources, Writing—review and editing; Siegfried Roth, Conceptualization, Formal analysis, Supervision, Funding acquisition, Writing—original draft, Project administration, Writing—review and editing

### Author ORCIDs
Matthew Alan Benton (iD) https://orcid.org/0000-0001-7953-0765
Matthias Pechmann (iD) http://orcid.org/0000-0002-0043-906X
Kristen A Panfilio (iD) http://orcid.org/0000-0002-6417-251X
Jeremy A Lynch (iD) http://orcid.org/0000-0001-7625-657X
Siegfried Roth (iD) https://orcid.org/0000-0001-5772-3558

### Decision letter and Author response
Decision letter https://doi.org/10.7554/eLife.47346.057
Author response https://doi.org/10.7554/eLife.47346.058

---

# Additional files

### Supplementary files
• Supplementary file 1. Primer List. Primers used to produce anti-sense RNA for in-situ hybridization and dsRNA for RNAi-mediated gene knockdown.
DOI: https://doi.org/10.7554/eLife.47346.048

• Transparent reporting form
DOI: https://doi.org/10.7554/eLife.47346.049

### Data availability
All data generated or analysed during this study are included in the manuscript and supporting files. Supplementary file 1 contains all primers used to amplify sequences for production of antisense RNA (ISH) and dsRNA (RNAi).

The following datasets were generated:

| Author(s) | Year | Dataset title | Dataset URL | Database and Identifier |
|---|---|---|---|---|
| Nadine Frey, Matthew Alan Benton, Rodrigo Nunes da Fonseca, Cornelia von Levetzow, Dominik Stappert, Muhammad Salim Hakeemi, Kai H Conrads, Matthias Pechmann, Kristen A Panfilio, Jeremy A Lynch, Siegfried Roth | 2019 | Gryllus bimaculatus mist mRNA, complete cds | https://www.ncbi.nlm.nih.gov/nuccore/MK962881 | NCBI, MK962881 |
| Nadine Frey, Matthew Alan Benton, Rodrigo Nunes da Fonseca, Cornelia von Levetzow, Dominik Stappert, Muhammad Salim Hakeemi, Kai H Conrads, Matthias Pechmann, Kristen A Panfilio, Jeremy A Lynch, Siegfried Roth | 2019 | Gryllus bimaculatus concertina mRNA, complete cds | https://www.ncbi.nlm.nih.gov/nuccore/MK962880.1/ | NCBI, MK962880.1 |
| Nadine Frey, Mat- | 2019 | Gryllus bimaculatus folded | https://www.ncbi.nlm. | NCBI, MK962882 |

thew Alan Benton, Rodrigo Nunes da Fonseca, Cornelia von Levetzow, Dominik Stappert, Muhammad Salim Hakeemi, Kai H Conrads, Matthias Pechmann, Kristen A Panfilio, Jeremy A Lynch, Siegfried Roth

gastrulation mRNA, complete cds

nih.gov/nuccore/MK962882/

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
