## [Decision Letter]

Thank you for submitting your article "Fog signaling has diverse roles in epithelial morphogenesis in insects" for consideration by *eLife*. Your article has been reviewed by K VijayRaghavan as the Senior Editor and Reviewing Editor, and two reviewers. The following individual involved in review of your submission has agreed to reveal their identity: Pavel Tomancak (Reviewer #2).

The reviewers have discussed the reviews with one another and the Reviewing Editor has drafted this decision to help you prepare a revised submission.

Summary:

Frey et al., present a comprehensive body of work investigating the evolutionary conservation of Fog signalling pathway in Tribolium castaneum. Fog signalling has been extensively implicated in tissue morphogenesis of *Drosophila*, in particular in connection to apical constriction during tissue folding at the gastrulation stage. The authors now show that many aspects of Fog signalling function are conserved in Tribolium where they are deployed in different yet related morphogenetic events. In addition, through analysis of hypomorphs (induced by embryonic RNAi) they uncover potential additional roles of the pathway in cellularization a possibly also serosa spreading. The paper is treasure trove of information, starting with detailed description of the expression patterns of key components of Fog signalling pathway in Tribolium, phenotypic description of parental knockdown phenotypes by live imaging, dissection of the genetic regulation of the pathway by the axial patterning systems in Tribolium and elucidation of additional roles of Fog in development unrelated to apical constriction. We find particularly intriguing the result concerning the defect in internalization of primordial germ cells in Tc-fog mutants.

The paper has a scholarly written introduction that gives a comprehensive overview of the rather extensive prior work on the subject in *Drosophila* and Tribolium. The results are presented with great clarity and are supported by high quality figures and rigorous statistical analysis with extensive repetition giving confidence in the reproducibility of the, often complex, phenotypes. The paper ends with interesting discussion of evolutionary implications of the comparative analysis of fog signalling in *Drosophila*, Tribolium and other insects. We are of the opinion that the paper is a valuable contribution to the nascent field of evo-devo of morphogenesis and combines several rarely before combined experimental approaches leading to interesting and far-reaching conclusions.

In its present form, however, the manuscript is largely descriptive and would benefit from a more detailed cell level analysis of the morphogenetic processes affected. Specifically, it would be desirable to demonstrate the cell biological phenotypes associated with defective cellularisation and serosal expansion in fog mutants in a little more detail to validate the statement that fog in Tribolium is not dedicated to "apical constriction". Also, it would be desirable to determine whether Tribolium fog can rescue *Drosophila* fog phenotypes or vice-versa.

The experiments suggested above and the others in the specific comments below are desirable, but the authors should focus on completing them in two months.

Some summary points come through from the reviews and discussions:

1) Better illustration of the apical constriction.

2) Rescue of Tribolium fog mutant with *Drosophila* Fog (by co-injection Tc-fog RNAi and Dm-fog; even if there is no rescue, it does not necessarily disqualify the paper).

3) Tone down the arguments regarding cells shape transitions and cell rearrangements in fog mutants (or show 3D/time-lapse data).

4) Consider discussion points suggested by both reviewers.

Essential revisions:

1) Given the relatively poor conservation between fog homologs in insects, and the differences in the effects they produce on insect morphogenesis (as described here), a question I have is whether the observed differences in fog function reflect differences in dose (same partners and downstream effectors but the extent of activation is different) or whether they reflect qualitative differences in fog function among insects (different components of the pathway and different downstream effects). We appreciate that this will be hard to tease apart and that the experiments with higher concentrations of embryonic RNAi (see comment 3) suggest there is dose dependence. Urbansky et al. looked at the effect of *Drosophila* fog overexpression in Chironomous embryos that do not significantly express or require fog in the mesoderm primordium and uncovered a difference in the mode of gastrulation (ingression vs invagination). Can Tribolium fog rescue *Drosophila* PMG/ventral furrow invagination defects? Or perhaps more relevant to this study, can *Drosophila* fog rescue cellularisation defects in Tribolium? A related question is whether the contribution of maternal fog to morphogenesis been examined in *Drosophila*?

2) Does fog KD also affect serosal cell heights? It would be useful to show the higher magnification images of serosal cells (images in Figure 6A', B') in orthogonal views to substantiate the effect on the cuboidal to squamous transition, as has been done for mesoderm internalization in Figure 4.

3) The blastoderm defects observed in the fog pathway KDs are very interesting. These are observed when higher concentrations of RNAi are injected. The movies reveal that cellularisation is delayed in all KDs and is followed by defects in integrity of the blastoderm. The phenotype appears later in mist KD embryos and is more drastic, although cellularisation itself seems more normal in these embryos compared to cta KD embryos that show earlier defects (Video 6). To strengthen the idea that they result from defective cellularisation, could the authors examine whether the furrows are as deep as in control at identical stages? Also, a nuclear marker/pH3 in immunostainings might help determine what the underlying cellular process affected might be.

4) It will be helpful to have a schematic in which the differences in gene regulation of fog pathway components are highlighted, and another that shows whether all components of the pathway (cta, mist and fog) are required for all effects.

5) The manuscript talks extensively about Fog signalling regulating apical constriction. However, Figure 4 panels A1–F' do not show apical constriction clearly. Given the known function of Fog in *Drosophila* apical constriction of mesodermal cells, it is certainly expected. In fact, we have independently seen clear evidence for apical constriction during primitive pit formation and mesoderm internalisation in our Tribolium studies using light sheet microscopy. Either remove explicit statements regarding apical constriction and referring to Figures or reproduce the data. Alternatively, we are willing to share our datasets with you. Please note, that this is NOT an attempt to get onto the paper. Given the central role apical constriction plays in Fog function and in the paper, one has to show it clearly, also in order to satisfy the tissue mechanics field.

6) We would like to bring your attention to a recent pre-print from Thomas Lecuit lab (Bailles et al.,). In this paper, the authors propose a mechanical morphogenetic wave operating to spread MyoII mediated cell invagination from an initial source established by Fog signalling. They present convincing evidence that the wave is not mediated by de novo fog transcription nor diffusion of Fog ligand. Since the situation is relatively similar in Tribolium (i.e. invagination occurring in the domain outside of fog expression) it may provide an alternative explanation for the phenotypes observed upon disruption of the different domains of fog expression.

Moreover, in both systems, Integrin mediate attachment of the blastoderm to the vitelline envelope has been implicated in the invagination process. We are sure that the authors are aware of the secondary posterior expression domain of Tc-inflated (outside of the anterior ventral attachment point, see Munster et al., Figure 3B) which could provide a rich source of discussion material.

We are am not sure how to interpret the phenotype of double knockdown of Tc-Toll and Tc-fog that shows the lack of primitive pit and posterior amniotic fold formation. Tc-Toll regulates Tc-fog expression domain and Tc-fog KD has the above-mentioned defects. Combining the two does not appear to show anything new.

7) The authors postulate that the expression of Fog in the serosa cells could indicate an additional function of Fog in these cells. To substantiate that claim, they analyze hypomorphic Tc-fog knockdowns that manage to close the serosa window. After the window closes, they analyse the size distribution of the serosa cells along the AP axis and conclude that the cells at the posterior are bigger compared to the anterior ones. The quantification is convincing (although we would like to know, how exactly was it done, what Fiji plugins were used, and how was the effect of partially covered cells at the periphery of the slice dealt with – simply more details in Materials and methods section) and one can even see that something is going on simply by eye. However, we are not sure that the interpretation is warranted. The authors discuss their prior observations on cell intercalations during serosa spreading, however, the link to the observed phenomenon is unclear. The authors mention technical difficulties with live imaging of cell intercalation during serosa window closure in the Tc-fog knockdown and yet this is precisely the experiment one would suggest to substantiate any connection between the phenotype and tissue fluidization. Perhaps the speculations should be moved to the Discussion section?

On the same topic. The presented data do not reveal anything about squamous, cuboidal or columnar nature of the various cells in the cellular blastoderm. High resolution 3D imaging allowing cross section and volume measurement would be needed (not an easy imaging experiment). Therefore, any discussion of epithelial cell shape transitions should be removed or at least toned down.

In the Discussion section, the authors should briefly explain, why are the phenotypes of parental and embryonic RNAi of Fog pathway components different to the point that the embryonic knockdown reveal qualitatively novel aspects of the phenotype. Intuitively, one would expect that the temporally earlier process of cellularization will be more affect by the parental RNAi.

---

## [Author Response]

Essential revisions:1) Given the relatively poor conservation between fog homologs in insects, and the differences in the effects they produce on insect morphogenesis (as described here), a question I have is whether the observed differences in fog function reflect differences in dose (same partners and downstream effectors but the extent of activation is different) or whether they reflect qualitative differences in fog function among insects (different components of the pathway and different downstream effects). We appreciate that this will be hard to tease apart and that the experiments with higher concentrations of embryonic RNAi (see comment 3) suggest there is dose dependence. Urbansky et al., looked at the effect of *Drosophila* fog overexpression in Chironomous embryos that do not significantly express or require fog in the mesoderm primordium and uncovered a difference in the mode of gastrulation (ingression vs invagination). Can Tribolium fog rescue *Drosophila* PMG/ventral furrow invagination defects? Or perhaps more relevant to this study, can *Drosophila* fog rescue cellularisation defects in Tribolium? A related question is whether the contribution of maternal fog to morphogenesis been examined in *Drosophila*?

These are very interesting questions. However, extensive additional work would be required to address these questions in a meaningful way.

We have previously found that overexpression of endogenous genes via microinjection is less efficient in Tribolium than in *Drosophila*, even to the extent of not working at all (n = 3 wildtype genes and one mutant form). Therefore, a negative result from microinjection of Dm-fog into Tc-fog RNAi embryos does not reveal anything meaningful. To control for this, one would need to first determine whether microinjection of Tc-fog into Tc-fog RNAi embryos rescues the phenotype. For this we would have to produce a Tc-fog sense construct with sufficient sequence changes preventing RNAi induced degradation. Even if this would work, the interpretation of the results would not be easy. Fog signalling in Tribolium has multiple additional functions compared to Fog signalling in Chironomus. We expect that the overexpression phenotypes might be very complex including alterations in cellularisation, serosa formation and posterior folding. The analysis of such phenotypes as a function of the amount of injected sense RNA are likely very interesting, but they would require an extensive new study.

As suggested by the reviewers, one could instead test whether Tc-fog rescues Dm-fog mutant embryos. However, over expression of Dm-fog in *Drosophila* also leads to complex phenotypes affecting all cells of the embryo (Morize et al., 1998). Thus, sense RNA injections would be hard to interpret. Transgenes would be required expressing the Tc-fog in the mesoderm, posterior gut or ubiquitously (e.g. hs-Tc-fog). Such experiments are definitely interesting but would again require a very careful analysis preferably using live and high resolution cellular imaging.

For the final question regarding maternal fog in *Drosophila*, the original articles describing fog and cta mutants (Sweeton et al., 1994, Parks and Wieschaus, 1991) examined maternal/zygotic mutants and reported no effects prior to gastrulation. We have included this information in the Discussion section.

2) Does fog KD also affect serosal cell heights? It would be useful to show the higher magnification images of serosal cells (images in Figure 6A', B') in orthogonal views to substantiate the effect on the cuboidal to squamous transition, as has been done for mesoderm internalization in Figure 4.

We have now added two movies (Video 6 for wt and Video 14 for Tc-fog KD), which demonstrate serosal cell flattening.

3) The blastoderm defects observed in the fog pathway KDs are very interesting. These are observed when higher concentrations of RNAi are injected. The movies reveal that cellularisation is delayed in all KDs and is followed by defects in integrity of the blastoderm. The phenotype appears later in mist KD embryos and is more drastic, although cellularisation itself seems more normal in these embryos compared to cta KD embryos that show earlier defects (Movie 6). To strengthen the idea that they result from defective cellularisation, could the authors examine whether the furrows are as deep as in control at identical stages? Also, a nuclear marker/pH3 in immunostainings might help determine what the underlying cellular process affected might be.

These phenotypes were highly variable, and this variability provided a considerable barrier to further analyses. For example, the observations of the reviewers based on Video 6 are not representative of the data as a whole. Indeed, aside from the overall weaker effect of fog KD compared with the other genes, we were unable to find consistent patterns in any phenotypic aspect that we assayed, including total number of cells, depth of membrane furrows, dynamics/position/size of the blastodermal ‘gaps’ or extent of blastoderm disintegration. We have expanded the description of these phenotypes to further emphasise the variability in results (Video 16).

With regards to the reviewers’ last point, in our previous experience characterising cellularisation defects (van der Zee et al., 2015) we found visualising nuclei was not informative

4) It will be helpful to have a schematic in which the differences in gene regulation of fog pathway components are highlighted, and another that shows whether all components of the pathway (cta, mist and fog) are required for all effects.

This is a great suggestion, and we have now added a schematic showing gene regulation (Figure 6—figure supplement 1). For the latter suggestion, KD of each component of the Fog pathway gave the same phenotypic effects and we have made this more explicit in the text.

5) The manuscript talks extensively about Fog signalling regulating apical constriction. However, Figure 4 panels A1–F' do not show apical constriction clearly. Given the known function of Fog in *Drosophila* apical constriction of mesodermal cells, it is certainly expected. In fact, we have independently seen clear evidence for apical constriction during primitive pit formation and mesoderm internalisation in our Tribolium studies using light sheet microscopy. Either remove explicit statements regarding apical constriction and referring to Figures or reproduce the data. Alternatively, we are willing to share our datasets with you. Please note, that this is NOT an attempt to get onto the paper. Given the central role apical constriction plays in Fog function and in the paper, one has to show it clearly, also in order to satisfy the tissue mechanics field.

We agree that describing the link between apical constriction and Fog signalling is important.

We have now added movies showing apical constriction at the posterior pole (Video 3, Video 4 and Video 5) and also showed a high magnification view of the mesoderm with cell-outlines highlighting apical constrictions (Figure 5—figure supplement 2). We have updated the methods to describe how the posterior pole imaging was performed.

6) We would like to bring your attention to a recent pre-print from Thomas Lecuit lab (Bailles A. et al.,). In this paper, the authors propose a mechanical morphogenetic wave operating to spread MyoII mediated cell invagination from an initial source established by Fog signalling. They present convincing evidence that the wave is not mediated by de novo fog transcription nor diffusion of Fog ligand. Since the situation is relatively similar in Tribolium (i.e. invagination occurring in the domain outside of fog expression) it may provide an alternative explanation for the phenotypes observed upon disruption of the different domains of fog expression.Moreover, in both systems, Integrin mediate attachment of the blastoderm to the vitelline envelope has been implicated in the invagination process. We are sure that the authors are aware of the secondary posterior expression domain of Tc-inflated (outside of the anterior ventral attachment point, see Munster et al., Figure 3B) which could provide a rich source of discussion material.We are not sure how to interpret the phenotype of double knockdown of Tc-Toll and Tc-fog that shows the lack of primitive pit and posterior amniotic fold formation. Tc-Toll regulates Tc-fog expression domain and Tc-fog KD has the above-mentioned defects. Combining the two does not appear to show anything new.

We thank the reviewers for highlighting the interesting discussion points and we have now included them in our manuscript. The Bailles et al., 2019 is now mentioned in the Introduction and later taken up in the Discussion section.

The double KD of fog and Toll was necessary for us to confirm that the symmetric invagination at the posterior of Toll KD embryos corresponds to a posterior amniotic fold of wt. In an earlier paper we had shown that Toll KD embryos largely lack the amnion/dorsal ectoderm (Nunes da Fonseca et al., 2008). Thus, it was not certain a priori how the folding events of these highly abnormal embryos correspond to wt folding. We have now made this point more clear in the text (subsection “The role of local *Tc-fog* expression for posterior folding”).

7) The authors postulate that the expression of Fog in the serosa cells could indicate an additional function of Fog in these cells. To substantiate that claim, they analyze hypomorphic Tc-fog knockdowns that manage to close the serosa window. After the window closes, they analyse the size distribution of the serosa cells along the AP axis and conclude that the cells at the posterior are bigger compared to the anterior ones. The quantification is convincing (although we would like to know, how exactly was it done, what Fiji plugins were used, and how was the effect of partially covered cells at the periphery of the slice dealt with – simply more details in Materials and methods) and one can even see that something is going on simply by eye. However, we are not sure that the interpretation is warranted. The authors discuss their prior observations on cell intercalations during serosa spreading, however, the link to the observed phenomenon is unclear. The authors mention technical difficulties with live imaging of cell intercalation during serosa window closure in the Tc-fog knockdown and yet this is precisely the experiment one would suggest to substantiate any connection between the phenotype and tissue fluidization. Perhaps the speculations should be moved to the Discussion section?On the same topic. The presented data do not reveal anything about squamous, cuboidal or columnar nature of the various cells in the cellular blastoderm. High resolution 3D imaging allowing cross section and volume measurement would be needed (not an easy imaging experiment). Therefore, any discussion of epithelial cell shape transitions should be removed or at least toned down.

We have now updated the methods to state that quantification was done manually with no Fiji plugins, and incomplete cells were excluded from analyses (Materials and methods section). With regards to serosa intercalation, we have now included descriptions and quantification of cell intercalation in the serosa in wildtype and knockdown embryos (new Figure 8 and Video 10, Video 11, Video 12 and Video 13). We found that the number of cell intercalations is higher in Tc-fog KD embryos compared to controls and discuss why this may be. We have updated our results, discussion and methods based on this new work. We note that in the last weeks, the reviewer Pavel Tomancak published a preprint describing the same process of serosa cell intercalation and we have mentioned it in our Discussion section.

With regards to cell shape changes, we have included new imaging as described in Point 2. Our live imaging of serosa spreading had to be performed with z-sampling of 2-3 µm between sections to enable many embryos to be injected/imaged simultaneously and provide the required statistical rigour. Therefore, the extremely attenuated nature of serosa cells (<1 µm at cell boundaries according to TEM from Hilbrant et al., 2016) coupled with the fluorescent signal from the overlying vitelline membrane and underlying yolk cell membrane prevented us from quantifying serosa cell heights. We agree that cell height (in addition to area) would be needed to quantify changes in volume, but cell area alone can be used as proxy for cuboidal to squamous transitions.

In the Discussion section, the authors should briefly explain, why are the phenotypes of parental and embryonic RNAi of Fog pathway components different to the point that the embryonic knockdown reveal qualitatively novel aspects of the phenotype. Intuitively, one would expect that the temporally earlier process of cellularization will be more affect by the parental RNAi.

This is a good topic for discussion, and we have now included it in the manuscript (Discussion section).